

# Summer-temperature evolution on the Kamchatka Peninsula, Russian Far East, during the past 20,000 years

Vera D. Meyer[1,2], Jens Hefter[1], Gerrit Lohmann[1], Ralf Tiedemann[1] and Gesine Mollenhauer[1,2,3]

[1] Alfred Wegener Institute Helmholtz Centre for Polar and Marine Research, Bremerhaven, 27570, Germany
[2] Department of Geosciences University of Bremen, Bremen, 28359, Germany
[3] MARUM- Centre for Environmental Sciences, University of Bremen, Bremen, 28359, Germany

*Correspondence to:* V. D. Meyer (vera.meyer@awi.de)

**Abstract.** Little is known about the climate evolution on the Kamchatka Peninsula during the last deglaciation as existing climate records do not reach beyond 12 ka BP. In this study, a summer-temperature record for the past 20 ka is presented. Branched Glycerol Dialkyl Glycerol Tetraethers, terrigenous biomarkers suitable for continental air temperature reconstructions, were analyzed in a sediment core from the western continental margin off Kamchatka/marginal Northwest Pacific (NW Pacific). The record reveals that summer temperatures on Kamchatka during the Last Glacial Maximum (LGM)
equaled modern. We suggest that strong southerly winds associated with a pronounced North Pacific High pressure system over the subarctic NW Pacific accounted for the warm conditions. A comparison with outputs from an Earth System Model reveals discrepancies between model and proxy-based reconstructions for the LGM-temperature and atmospheric circulation in the NW Pacific realm. The deglacial temperature development is characterized by abrupt millennial-scale temperature oscillations. The Bølling/Allerød warm-phase and the Younger Dryas cold-spell are pronounced events, providing evidence
for a strong impact of North-Atlantic climate variability on temperature development in southeastern Siberia. Summer insolation and teleconnections with the North Atlantic determine the long-term temperature development during the Holocene.

**Key words**: CBT/MBT, summer temperature, Northwest Pacific, deglaciation, atmospheric circulation

## 1.   Introduction

The Kamchatka Peninsula is attached to Siberia and protrudes into the North Pacific Ocean separating the Sea of Okhotsk from the Northwest Pacific (NW Pacific) and the Bering Sea (Fig. 1a). The Peninsula is a remote part of western Beringia. "Beringia" extends from the Lena River in Northeast Russia to the lower Mackenzie River in Canada (Fig. 1a, Hopkins et al.,





1982). During Pleistocene sea-level low-stands the Bering Land Bridge (BLB) linked Eastern and Western Beringia as the

Chukchi and Bering Shelves became exposed (Fig. 1a). Kamchatka is one of the least studied areas of Beringia since the

available terrestrial climate archives, such as peat sections or lake sediments, do not reach beyond 12 ka BP (e.g. Dirksen et

al., 2013, 2015; Nazarova et al., 2013a; Hoff et al. 2015; Klimaschewski et al., 2015; Self et al., 2015; Solovieva et al., 2015)

and the climatic conditions during the LGM and the deglaciation are poorly understood. However, the climatic history of

Kamchatka may provide important insights into the deglacial development of regional atmospheric and oceanic circulation,

since the Holocene climate evolution largely responds to those regional forcing mechanisms (Nazarova et al., 2013a; Brooks

et al., 2015; Hammarlund et al., 2015; Self et al., 2015) next to global or supra-regional climate drivers, e.g. summer insolation

(Savoskul, 1999; Dirksen et al., 2013; Brooks et al., 2015; Self et al., 2015). Particularly, information about atmospheric and

oceanic circulation in the Northwest Pacific (NW Pacific) realm is important to confirm outputs from climate models.

The investigation of deglacial climate change on Kamchatka may also contribute to the understanding of the spatial dimension

of atmospheric teleconnections with abrupt climate change in the North Atlantic (N-Atlantic). The majority of sea surface

temperature records from the subarctic NW Pacific and the marginal seas mirror the N-Atlantic climate oscillations (e.g.

Caissie et al., 2010; Max et al., 2012; Meyer et al., submitted) suggesting that atmospheric teleconnections with the North

Atlantic controlled deglacial temperature development in the N-Pacific realm (Max et al., 2012; Meyer et al., submitted).

However, climate records from Siberia and Alaska provide an ambiguous picture concerning the sensitivity of Beringia to

climate oscillations in the N Atlantic. Some studies in Siberia and interior Alaska found patterns similar to the N-Atlantic

climate variability, including a Bølling/Allerød (B/A)-equivalent warm-phase and a subsequent climatic reversal during the

Younger Dryas (YD; Anderson et al., 1990; Andreev et al., 1997; Pisaric et al., 2001; Bigelow and Edwards, 2001; Brubaker

et al., 2001; Anderson et al., 2002; Meyer et al., 2010; Anderson and Lozhkin, 2015), while other Alaskan and east Siberian

records show progressive warming during the postglacial climate amelioration, without a YD-cold spell (Lozhkin et al., 1993,

2001; Anderson et al., 1996, 2002; Lozhkin and Anderson, 1996; Nowaczyk et al., 2002; Anderson et al., 2003; Nolan et al.,

2003; Kokorowski et al., 2008a,b; Kurek et al., 2009). As pointed out by Kokorowski et al. (2008a,b) this may attest to regional

differences or to uncertainties in chronologies. Therefore, further deglacial climate records with high resolution are necessary.

This particularly applies for easternmost Siberia, since most deglacial records are obtained from sites west of 150°N and north of 65°N (Kokorowski et al., 2008a).

In this study, we analyzed branched glycerol dialkyl glycerol tetraethers (brGDGTs), terrigenous biomarkers as recorders of continental temperature (Weijers et al., 2006a, 2007), in a marine sediment core retrieved at the eastern continental margin off Kamchatka/NW Pacific (site SO201-2-12KL, NW Pacific, Fig. 1a, b). We present a continuous, quantitative record of summer-temperature on Kamchatka for the past 20 ka. The impact of global climate drivers, N-Atlantic climate change, and regional atmospheric/oceanic circulation is investigated. The record reveals new aspects of LGM atmospheric circulation in the NW

Pacific-realm, which are compared to an Earth System Model (ESM), and provides new insights into the interplay of global and regional climate drivers in the south-eastern edge of western Beringia since the LGM.

## 2.    Regional Setting

The Kamchatka Peninsula is situated south of the Koryak Uplands in Siberia. It is characterized by strong variations in relief with lowlands in the coastal areas (Western Lowlands; Eastern Coast) and mountain ranges further inland (Fig. 1b). The

mountain ranges, the Sredinny and the Eastern Ranges, encircle the lowlands of the Central Kamchatka Depression (CKD; Fig. 1b). The CKD is the largest watershed of the Peninsula and is drained by the Kamchatka River, the largest river of Kamchatka. The river discharges into the Bering Sea near 56°N (Fig. 1b). The climate is determined by marine influences from the surrounding seas, by the East Asian continent, and by the interplay of the major atmospheric pressure systems over NE-Asia and the North Pacific (e.g. Mock et al., 1998; Glebova et al., 2009). In general the climate is classified as sub-arctic

maritime (Dirksen et al., 2013). The winters are characterized by cold and relatively continental conditions since northerly winds prevail over Kamchatka which are mainly associated with the Aleutian Low over the N Pacific and the Siberian High over the continent (Mock et al., 1998). In summer, Kamchatka experiences warm maritime conditions owing to the East Asian Low over the continent and the North Pacific High (NPH) over the N-Pacific (Mock et al., 1998). Furthermore, there are the influences of the East Asian Trough (EAT) which has its average position over the northern shelves of central Beringia, as

well as the influences of the westerly Jet and the associated polar front (Mock et al., 1998). Variations in the position and strength of the EAT affect precipitation and temperature over Beringia and can cause climatic contrasts between Siberia and

Alaska (Mock et al., 1998 and references therein). With respect to Kamchatka westerly to northwesterly winds associated with the Jetstream and the EAT form a source of continental air masses from Siberia/East Asia (Mock et al., 1998).

The mountainous terrain with strongly variable relief results in pronounced climatic diversity on the Peninsula (Fig. 1b). The
coastal areas, the western Lowlands and the Eastern Coast, are dominated by marine influences. In the coastal areas, summers are cool and wet and winters are relatively mild. Precipitation is high along the coast and in the mountains throughout the year (Kondratyuk, 1974; Dirksen et al., 2013). Being protected from marine influences by the mountain ranges the CKD has more continental conditions with less precipitation and a larger annual temperature range than in the coastal areas (Ivanov, 2002; Dirksen et al., 2013, Kondratyuk, 1974; Jones and Solomina, 2015). Averaged for the entire Peninsula mean temperatures
range from -8 to -26°C in January and from 10 to 15°C in July (Ivanov, 2002).

## 3. Material and Methods

### 3.1. Core material and chronology

Within a joint German/Russian research program (KALMAR Leg 2) core SO201-2-12KL (Fig. 1a, b) was recovered with a piston-corer device during cruise R/V SONNE SO201 in 2009 (Dullo et al., 2009). The core material was stored at 4°C prior
to sample preparation. Age control is based on accelerator mass spectrometry (AMS) radiocarbon dating of planktic foraminifera (*Neogloboquadrina pachyderma* sin.) as well as on core-to-core correlations of high-resolution spectrophotometric (color b*) and X-ray fluorescence (XRF) data. For detailed information and AMS-$^{14}$C results, see Max et al. (2012).

### 3.2. Lipid extraction

For GDGT analyses, freeze-dried and homogenized sediment samples (approx. 5 g) were extracted with dichloromethane : methanol (DCM:MeOH, 9:1 v/v) using accelerated solvent extraction (ASE). Prior to extraction, 10 µg of a $C_{46}$-GDGT internal standard was added to each sample. The extraction was conducted on a "Dionex ASE 200"-device and was performed in three cycles, each of them lasting for five minutes. During the extraction cycles the temperature was maintained at 100°C and the

pressure at 1000 psi. After drying with a rotary-evaporator, extracts were hydrolyzed with 0.1N potassium hydroxide (KOH)

in MeOH:$H_2O$ 9:1 (v/v) to separate carbonic acids from neutral compound classes. After the hydrolyzation, neutral compounds

such as hydrocarbons, ketones, alcohols and GDGTs were extracted with *n*-hexane, from the saponified solution. Dissolved in

*n*-hexane the neutral compound-classes were separated using silica gel columns. Columns were built with Pasteur pipettes

(6 mm diameter) which were filled with deactivated $SiO_2$ (mesh size 60, filling height 4 cm). After having eluted a less polar

fraction with *n*-hexane, a polar fraction, containing the GDGTs, was eluted with DCM:MeOH (1:1 v/v). Dried polar fractions

were dissolved in *n*-hexane:isopropanol (99:1, v/v) and were filtered through PTFE syringe filters (4 mm diameter, 0.45 μm

pore size). Afterwards, samples were brought to a concentration of 2 μg/μl in order to prepare them for GDGT analysis.

### 3.3.  GDGT analysis

GDGTs were analyzed by High Performance Liquid Chromatography (HPLC) and a single quadrupole mass spectrometer

(MS). The systems were coupled via an atmospheric pressure chemical ionization (APCI) interface. The applied method was

slightly modified from Hopmans et al. (2000). Analyses were performed on an Agilent 1200 series HPLC system and an

Agilent 6120 MSD. Separation of the individual GDGTs was performed on a Prevail Cyano column (Grace, 3 μm, 150 mm x

2.1 mm) which was maintained at 30°C. After sample injection (20 μL) and 5 min isocratic elution with solvent A (hexane)

and B (hexane with 5% isopropanol) at a mixing ratio of 80:20, the proportion of B was increased linearly to 36% within 40

min. The eluent flow was 0.2 ml/min. After each sample, the column was cleaned by back-flushing with 100% solvent B (8

min) and re-equilibrated with solvent A (12 min, flow 0.4 ml/min). GDGTs were detected using positive-ion APCI-MS and

selective ion monitoring (SIM) of their (M+H)$^+$ ions (Schouten et al., 2007). APCI spray-chamber conditions were set as

follows: nebulizer pressure 50 psi, vaporizer temperature 350 °C, $N_2$ drying gas flow 5 l/min and 350 °C, capillary voltage

(ion transfer tube) -4 kV and corona current +5 μA. The MS-detector was set in SIM-mode detecting the following (M+H)$^+$

ions with a dwell time of 67 ms per ion: *m/z* 1292.3 (GDGT 4 + 4´ / crenarcheol + regio-isomer), 1050 (GDGT IIIa), 1048

(GDGT IIIb), 1046 (GDGT IIIc), 1036 (GDGT IIa), 1034 (GDGT IIb), 1032 (GDGT IIc), 1022 (GDGT Ia), 1020 (GDGT Ib),

1018 (GDGT Ic) and 744 ($C_{46}$-internal standard).

GDGTs were quantified by peak-integration and the obtained response factor from the $C_{46}$ -standard. Concentrations were normalized to the dry weight (dw) of the extracted sediment and to total organic carbon contents (TOC). It has to be noted that the quantification should only be regarded as semi-quantitative because individual relative response factors between the $C_{46}$-

standard and the different GDGTs could not be determined due to the lack of appropriate standards. Fractional abundances of single GDGTs were calculated relative to the total abundance of the all nine brGDGTs. The standard deviation was determined from repeated measurements of a standard sediment and resulted in an uncertainty of 9 % for the concentration of the sum of all nine brGDGT (ΣbrGDGT).

### 3.4.    Temperature determination

The Cyclysation of Branched Tetraether index (CBT) and Methylation of Branched Tetraether index (MBT) were introduced as proxies for soil-pH (CBT) and mean annual air temperature (MAT, CBT/MBT) by Weijers et al. (2007). The CBT-index was calculated after Weijers et al. (2007). For calculating the MBT-index we used a modified version of the original index, the MBT' which excludes GDGTs IIIb and IIIc, and was introduced by Peterse et al. (2012). From repeated measurements the standard deviation for CBT and MBT' were determined as 0.01 and 0.04, respectively. CBT and MBT'-values were converted

into temperature using the global-soil dataset calibration by Peterse et al. (2012). The residual standard mean error of this calibration is 5°C (Peterse et al., 2012). The standard deviation of CBT and MBT' translates into an uncertainty of max. 0.1°C.

Although terrestrial soils are supposed to be the main source of branched GDGTs (Weijers et al., 2007) brGDGT can also be produced in-situ in marine water systems (Peterse et al., 2009; Zhu et al., 2011; Zell et al., 2014) as well as in fresh water environments such as rivers or lakes (Tierney 2010; Zell et al., 2013; De Jonge et al., 2014; Dong et al., 2015). As in-situ

production can bias temperature reconstructions, particularly in marine settings where the input of terrigenous GDGTs is low (Weijers et al., 2006b; Peterse et al., 2009, 2014; DeJonge et al., 2014), the contribution of brGDGTs to the marine sediments needs to be estimated prior to any paleoclimatic interpretation of CBT/MBT'-derived temperatures. A common means to estimate the relative input of marine and terrestrial GDGTs is the BIT-index (Branched and isoprenoid tetraether index) which quantifies the relative contribution of the marine-derived Crenarchaeol and terrigenous brGDGTs (Hopmans et al., 2004). BIT-

values were adopted from Meyer et al. (submitted) who worked on the same sample used in this present study.

### 3.5.    Climate simulations with the Earth System Model COSMOS

In order to compare inferences for atmospheric circulation during the summer months to computer model outputs, model

simulations for the glacial climate were performed with the Earth System model COSMOS for pre-industrial (Wei et al.,

2012) and glacial conditions (Zhang et al., 2013). The model configuration includes the atmosphere component ECHAM5 at

T31 resolution (~3.75°) with 19 vertical layers (Roeckner et al., 2006), complemented by a land-surface scheme including

dynamical vegetation (Brovkin et al., 2009). The ocean component MPI-OM, including the dynamics of sea ice formulated

using viscous-plastic rheology, has an average horizontal resolution of 3º×1.8° with 40 uneven vertical layers (Marsland et

al, 2003). The performance of this climate model was evaluated for the Holocene (Wei and Lohmann, 2012; Lohmann et al.,

2013), the last millennium (Jungclaus et al., 2006), glacial millennial-scale variability (Gong et al., 2013; Weber et al., 2014;

Zhang et al., 2014), and warm climates in the Miocene (Knorr and Lohmann, 2014) and Pliocene (Stepanek and Lohmann,

2012).

The climate model was integrated for 3000 model years and provides monthly output. Here, anomalies in sea-level pressure

(SLP), wind directions (1000 hPa level) and surface air temperature (SAT) between the LGM and pre-industrial conditions

were analyzed for the boreal summer season - June, July and August (JJA). All produced Figures show climatological mean

characteristics averaged over a period of 100 years at the end of each simulation.

### 4.    Results

#### 4.1. Concentrations and fractional abundance of brGDGT

The summed concentration of all nine brGDGTs (ΣbrGDGT) is shown in Figure. 2a. The concentration of ΣbrGDGTs vary

between 40 and 160 ng/g dw throughout the record. Ranging between 60-80 ng/g dw, they are lowest during the LGM and the

late Holocene. During the deglaciation and the early Holocene (17-8 ka BP) lowest values are approx. 80 ng/g dw, except for

two peaks at 15-16 ka BP and 12-13 ka BP, respectively, where concentrations reach 160 ng/g dw (Fig. 2a).

The fractional abundance of all nine brGDGTs, calculated relative to the ΣbrGDGT, is shown in Figure. 3. All samples are characterized by a similar pattern. The composition of the brGDGT assemblage is dominated by brGDGTs without cyclopentyl moieties which together account for 60-80% of the total GDGT-assemblage (GDGT Ia, IIa, IIIa; Fig. 3). GDGTs with a higher degree of methylation are more abundant than lesser methylated ones. In 83 out of 90 samples GDGT IIIa is the most prominent GDGT accounting for 22-37% of the total GDGT distribution. It is closely followed by GDGT IIa with 16-29% and GDGT Ia which accounts for 14-23% of the total GDGT distribution. As for GDGTs containing cyclopentyl moieties, GDGT IIb is most abundant accounting for 9-16% of the total GDGT assemblage. GDGT IIc, Ib, Ic, IIIb and IIIc are less abundant reaching 2-6%, 3-7%, 1-3%, 2-4%, and 1-2%. In one outlying sample GDGT IIc accounts for 24% (Fig. 3).

### 4.2. Temperature development over the past 20 ka

The CBT/MBT'-derived temperatures are plotted in Fig. 2b. During the late Holocene (approx. 1 ka BP), the reconstructed temperature is 7.5°C. Interestingly, glacial temperatures (between 20-18 ka) are the same (Fig. 2b). At 18 ka temperature drops by about 1.5°C. At 16 ka temperature jumps back to the glacial level. As this increase is based on one single data point, it cannot be excluded that this warming is an artifact resulting from an outlier. Deeming the data-point an outlier, temperature increases progressively until the onset of the Bølling/Allerød at approx. 14.6 ka BP, where it abruptly jumps back to the glacial and Holocene level of 7.5°C (Fig. 2e). Between 14.6 and 13 ka, temperature progressively decreases about 1-0.5°C. During the Younger Dryas (YD) temperature abruptly decreases by about 2°C (at approx. 13 ka BP) and remains cold until 12 ka BP (Fig. 2b). With approx. 4.5°C the YD is the coldest episode during the Glacial-Holocene transition. The cold spell is followed by a sharp temperature increase of approx. 3°C at the onset of the Preboreal (PB)/early Holocene (Fig. 2b). After the abrupt temperature increase into the PB temperature progressively increases culminating in a Mid-Holocene Thermal Maximum (HTM) between 8.0-4.0 ka BP. With 8°C being reached between 6 and 4 ka BP, the mid-Holocene is the warmest episode since the Last Glacial Maximum (LGM). At 4 ka BP a cooling trend initiates and temperature decreases by about 0.5°C (Fig. 2b). Compared to the deglacial temperature variations the Holocene variability is relatively small.

### 4.3. LGM-climate simulation with COSMOS

#### 4.3.1. Sea-level pressure and wind patterns

Model-simulations for SLP (JJA) are shown Fig. 4a. The LGM-simulation is characterized by strong positive anomalies in sea-level pressure (SLP) over the American Continent (Fig. 4a). Positive SLP-anomalies also occur over the Arctic Ocean. Negative SLP anomalies occur south of 50°N and are centered over the NW Pacific and East Asia, but are also observed in a few grid-cells over the central and NE Pacific and over the Sea of Okhotsk. In the Bering Sea, the northern N-Pacific (north of 50°N) and Beringia SLP does not change significantly relative to present.

The strong positive SLP-anomalies over North America are associated with pronounced anticyclonic anomalies in the wind directions, which expand to the Chukchi-Sea and to the formerly exposed BLB (Fig. 4a). Over western Beringia as well as the adjacent Arctic Ocean small northerly anomalies are present. Between 100°E and 110°E pronounced anticyclonic anomalies are present over Russia. Over Kamchatka and the adjacent East Siberian Coast small northerly anomalies occur. The western Bering Sea is characterized by easterly anomalies. Over the NW Pacific anomalies are small and show now general pattern. In the NE Pacific relatively strong westerly to southwesterly anomalies are present.

#### 4.3.2. Surface air temperature

Model simulations for SAT (JJA) are shown in Fig. 4b. The model predicts widespread negative surface air temperature (SAT)-anomalies over Beringia, East Asia, North America, the Arctic Ocean and the entire N-Pacific (Fig. 4b). However, in small parts of the formerly exposed BLB slightly warmer-than-present conditions are simulated. On the arctic shelf a small band where temperature may equal the PI-conditions as the SAT anomaly falls in the window of -1 to +1°C, occurs. The temperature anomalies are strongest over North America where they reach -17°C. Over western Beringia the SAT anomaly increases from east to west with SAT ranging between -1 and -5 over East Siberia and between -5 and -9 further west. Over the N-Pacific SAT anomalies are smaller than over western Beringia and range between -1 and -5°C. SAT anomalies are smallest in the Bering Sea and along the eastern coast of Kamchatka. Over the Peninsula itself, the majority of grid-cells indicate a negative



anomaly (-3 to -5°C). In the northern part and over the adjacent Bering Sea the SAT anomalies are very small within the

window of -1 to +1°C (Fig. 4b).

## 5.    Discussion

### 5.1.    Sources of brGDGT and implications for CBT/MBT'-derived temperatures

Considering that brGDGT are thought to be synthesized by terrestrial bacteria which thrive in peats and soils (e. g. Weijers et

al., 2006b) it is most likely that the major origin of brGDGT in the marine sediments of the Bering Sea/NW Pacific would be

the Kamchatka Peninsula. However, BIT-values from core 12KL range between 0.08 and 0.2 (Meyer et al., submitted)

throughout the entire record, indicating that marine derived GDGT dominate the total GDGT composition and that terrigenous

input is low (Fig. 2c). Since a bias from in-situ production is particularly eminent in marine settings where terrigenous input

is low (e.g. Weijers et al., 2006b; Peterse et al., 2009; Zhu et al., 2011), non-soil derived brGDGTs potentially have a

considerable effect on the temperature reconstruction at site 12KL. However, the concentrations of ΣbrGDGT show strong

similarities with the trend of Titanium/Calcium ratios (Ti/Ca-ratios, Fig. 2d) from core 12KL (XRF-data from Max et al.

(2012)). Reflecting the proportion of terrigenous and marine derived inorganic components of the sediment, Ti/Ca-ratios can

be used as an estimator of terrigenous input. With relatively high values at 15.5 and 12 ka BP, and minima at 14 and 11 ka BP.

As intervals of relatively high/low terrigenous input (as suggested by Ti/Ca) coincide with relatively high/low ΣbrGDGT-

concentrations brGDGTs seem to be terrigenous (Fig. 2b, d). Moreover, the distribution of the brGDGTs the samples from site

12KL resemble the GDGT composition described for soils world-wide (Weijers et al., 2007; Blaga et al., 2010) as GDGT Ia,

IIa and IIIa dominate over GDGTs with cyclopentyl moieties (e.g. Ib, IIb) accounting for 60-80% of the total brGDGT

assemblage (Fig. 3). By contrast, in areas where GDGTs are thought to be produced in-situ, the GDGT compositions were

dominated by GDGTs containing cyclopentyl moieties (Peterse et al., 2009; Zell et al., 2014). Thus, brGDGT seem to be soil-

derived and a bias from in-situ production is unlikely. We also exclude changes in the source of brGDGTs through time because

the relative abundance of the brGDGTs is similar in all samples indicating that the source of brGDGTs remained constant

throughout the past 20 ka (Fig. 3). We consider the catchment of the Kamchatka River (CKD and inner flanks of the mountains)

and the Eastern Coast as the likely sources of brGDGTs deposited in the marine sediments at the core site since the Kamchatka



River and several small rivers draining the Eastern Coast discharge into the western Bering Sea. Flowing southward along

Kamchatka, the East Kamchatka Current would carry the load of the Kamchatka River to site 12KL (Fig. 1b)

Although the CBT/MBT-paleothermometre has been suggested to generally record mean annual air temperatures (Weijers et

al., 2007) it is assumed to be biased to the summer months/ice-free season in high latitudes (Rueda et al., 2009, Shannahan et

al., 2013; Peterse et al., 2014). According to Klyuchi climate station (for location see Fig. 1b), mean annual air temperatures

in the northern CKD are approx. -0.5°C (http://en.climate-data.org/location/284590/). The CBT/MBT'-derived temperatures

for the core-top/late Holocene (7.5°C; Fig. 2) exceed the annual mean by approx. 8°C and are similar to mean air temperatures

from the ice-free season (Mai-October) at Klyuchi (approx. 9°C). Therefore, they are interpreted as summer temperature and

will be referred to as "Mean Air Temperature of the ice-free season" ($MAT_{ifs}$) henceforth.

### 5.2. Temperature evolution over the past 20 ka

### 5.2.1. The LGM (20-18 ka) – warm summers and the regional context

The finding that LGM and Holocene $MAT_{ifs}$ are equal contrasts with the general understanding of the glacial climate according

to which the extratropics were significantly colder than today, as documented by several proxy-based temperature

reconstructions (e.g. MARGO compilation, Kageyama et al., 2006; Waelbroeck et al., 2009) and computer model simulations

(e.g. Kutzbach et al., 1998; Kageyama et al., 2006; Kim et al., 2008; Alder and Hostetler, 2015). The general cooling tendency

is thought to result from low summer insolation, reduced carbon-dioxide concentrations in the atmosphere and extensive

continental ice caps (Berger and Loutre 1991; Monnin et al., 2001; Kageyama et al., 2006, Shakun et al., 2012). Therefore,

one may expect that the Kamchatka Peninsula would experience a glacial-interglacial warming trend. As $MAT_{ifs}$ deviates from

the trends in $CO_{2atm}$ and insolation (Fig. 2b, e, f) regional climate drivers may have overprinted the effects of $CO_{2atm}$ and

summer insolation. Interestingly, several studies investigating climate in Beringia based on pollen and beetle-assemblages

indicate that in NE Siberia and the formerly exposed BLB (catchments of the Lena, Kolyma and Indigirka Rivers, Ayon Island,

Anadyr Lowlands, Lake El'Gygytgen, Seward Peninsula, Fig. 4c) summers during the LGM were as warm as at present or

were even warmer (Fig. 4c; Elias et al., 1996, 1997; Elias, 2001; Alfimov and Berman, 2001; Kienast, 2002; Kienast et al.,

2005; Sher et al., 2005; Berman et al., 2011). Only a few pollen and insect-data from Markovo, Lakes Jack London and

El'Gygytgyn (Fig. 1a), point to colder-than-present conditions (Fig. 4c; Lozhkin et al., 1993; Alfimov and Bermann, 2001; Lozhkin et al., 2007; Pitul'ko et al., 2007). The fairly large number of sites indicating warm summers in Siberia suggests that a thermal anomaly was widespread over western/central Beringia (Fig. 4c) and extended to Kamchatka. The thermal anomaly did probably not extend to eastern Beringia as insect-data as well as pollen consistently point to summer cooling of up to 4°C (Fig. 4c; e.g. Mathews and Telka, 1997; Elias, 2001; Kurek et al., 2009).

### 5.2.2.    Controls on MAT$_{ifs}$

The warm Siberian summers were attributed to increased continentality, which would arise from the exposure of the extensive Siberian and Chukchi shelves at times of lowered sea-level (Fig. 1a; e.g. Guthrie, 2001; Kienast et al., 2005; Berman et al., 2011). The greater northward extent of the Beringian landmass (approx. +800 km relative to today) would have minimized maritime influences from the cold Siberian and Chukchi Seas (Guthrie, 2001; Alfimov and Berman, 2001; Kienast et al., 2005; Sher et al., 2005; Berman et al., 2011). Increased seasonal contrasts resulting in warmer summers and colder winters would have been the result (e.g. Guthrie, 2001; Kienast et al., 2005). Winter cooling in Siberia (relative to modern) is indicated by ice-wedge data (Meyer et al., 2002) from Bykovski Peninsula (Fig. 1a). Also, the presence of stronger-than-present sea-ice cover in the Bering Sea (Caissie et al., 2010; Smirnova et al., 2015) points to cold winters in Siberia and Kamchatka during the LGM. However, for Kamchatka it is unlikely that the thermal anomaly and an increased seasonal contrast were a direct result from lowered sea-level as the bathymetry around the Peninsula is relatively steep and the exposed shelf area was very small. (Fig. 1a, b). Thus, other climate drivers were likely responsible for the relatively warm summer conditions. Potential mechanisms are changes in oceanic or atmospheric circulation.

Intriguingly, alkenone-based SST reconstructions from the Sea of Okhotsk indicate that glacial SST were slightly warmer than today or equal to modern conditions (Seki et al., 2004, 2009; Harada et al., 2004, 2012; Fig. 4c). However, these records are considered to be biased by seasonal variations in the alkenone production rather than to reflect real temperature anomalies (Seki et al., 2004, 2009; Harada et al., 2004, 2012). This seems to be supported by a few TEX$^{L}_{86}$-based SST reconstruction from the Sea of Okhotsk suggesting that LGM SST were approx. 5°C colder than at present (Seki et al. 2009; 2014). In this light, a climatic relation between alkenone-based SST and MAT$_{ifs}$ seems very unlikely. Interestingly, LGM-SST in the

subarctic NW Pacific (site 12KL) were only 1°C lower than at present (Fig. 2 h), a relatively small temperature difference

compared to other SST records from the NW Pacific and its marginal seas which suggest a cooling of 4-5°C on average (e.g.

Seki et al., 2009; 2014; Harada et al., 2012, Meyer et al., submitted). The relatively warm SST at site 12KL were explained by

a stronger-than-present influence of the Alaskan Stream (Fig. 1a) in the marginal NW Pacific (Meyer et al., submitted). Such

warm SST may have supported the establishment of warm conditions on Kamchatka. However, it is unlikely, that the

temperature development on Kamchatka was fully controlled by oceanic influences since this would probably cause a similar

temperature reduction as in the SST record of site 12KL.

If oceanic circulation alone is unlikely to have caused the warm temperatures on Kamchatka, atmospheric circulation may

have exerted a strong control on glacial summer temperatures in the region. In terms of atmospheric circulation the summer

climate of the Kamchatka is largely determined by the strength and position of the North Pacific High (NPH) over the N Pacific

(Mock et al., 1998). As the southerly flow at the southwestern edge of the NPH brings warm and moist air masses to Kamchatka

summers on the Peninsula become warmer when the NPH and the associated warm southerly flow increase in strength (Mock

et al., 1998). This modern analogue suggests that the LGM-NPH over the subarctic NW was stronger than today and the

resulting warming effect may have balanced the cooling effects of $CO_{2atm}$ and insolation. This atmospheric pattern could be

explained by an increased thermal gradient between western/central Beringia and the N-Pacific Ocean. While warm summers

were widespread in western Beringia, the majority of sea surface temperature (SST) records from the open N Pacific and the

Bering Sea indicate colder conditions during the LGM (Fig. 4A; deVernal and Pedersen, 1997; Seki et al., 2009, 2014; Kiefer

and Kienast, 2005; Harada et al., 2004; 2012; Maier et al., 2015; Meyer et al., submitted). Under the assumption that alkenone-

based reconstructions of LGM SST in the Sea of Okhotsk are biased, also the Sea of Okhotsk may have been 4-5°C colder

than at present as suggested by $TEX^L_{86}$-based SST reconstruction (Seki et al. 2009; 2014). An increased thermal gradient

between the subarctic N Pacific and western Beringia would translate into an increased pressure gradient between the low-

pressure over western Beringia and the high pressure over the subarctic NW Pacific, and in response the southerly flow over

the Kamchatka would have intensified relative to today. (Fig. 4c).



### 5.2.2.1.  Comparison to the COSMOS-simulations

These inferences contrast with results from the climate simulations with COSMOS. For JJA the model predicts a decrease in

SLP over the NW-Pacific suggesting that the southerly flow at the western edge of the NPH was reduced rather than

strengthened (Fig. 4a). The weakening of the southerly flow is also discernable in the anomaly of the major wind-patterns

over the NW Pacific (Fig. 4a) as a small northerly anomaly occurs north of Kamchatka (Fig. 4a). The weakening of the NPH

is agreement with several other General Circulation Model (GCM) outputs, which consistently predict a reduction in SLP

over the N-Pacific (Kutzbach and Wright, 1985; Bartlein et al., 1998; Dong and Valdes, 1998; Vetteoretti et al., 2000;

Yanase and Abe-Ouchi, 2007; Alder and Hostetler, 2015). According to the climate synopsis by Mock et al (1998) a

northerly anomaly would have caused summer cooling on Kamchatka. It has been suggested that a pronounced positive SLP-

anomaly and a persistent anticyclone over the American continent resulted in reduced SLP over the Western North Pacific

(Yanase and Abe Ouchi, 2010). The positive SLP-anomaly and the strong anticyclonic tendencies are clearly present in the

COSMOS simulation of SLP and wind-patterns (Fig. 4a) and were also simulated by several other GCMs (e.g. Yanase and

Abe-Ouchi, 2007; 2010; Alder and Hostetler, 2015). Its development was attributed to the presence of extensive ice sheets

on the American continent (Yanase and Abe-Ouchi, 2010), which would have caused severe cooling of the overlying

atmosphere. Considering the consistency of different GCMs, the anticyclonic anomalies over North America as well as

resulting cyclonic anomalies over the N-Pacific seem to be a robust feature of the glacial atmospheric circulation. As this

contrast with the inferences made from the $MAT_{ifs}$-record, one may speculate that the effect of the ice-caps on the NPH

mainly influenced the NE Pacific and that a strengthened anticyclone (as suggested in sec. 5.2.2) was restricted to the

subarctic NW Pacific. In other words, the NPH may have shifted westward in response to the presence of a strong

anticyclonic anomaly over the LIS.

The COSMOS-simulation also contrasts with the temperature patterns in western Beringia suggested by proxy-based climate

reconstructions (see. Sec. 5.1) as summers were simulated to be colder than at present on Kamchatka and in Siberia (Fig. 4b).

However, in small parts of the formerly exposed BLB and the arctic shelves temperatures level or exceed PI-conditions (Fig.

4b). These positive anomalies in the model are probably associated with the dominant anticyclonic flow over North America



and the associated easterly to southeasterly winds over south-Alaska and the BLB (Fig. 4b). The exposure of the Siberian Shelf

may also have an effect. However, these anomalies are restricted to a relatively small area and are not comparable with the

widespread warming tendencies over Siberia, which are visible in the proxy-compilation (Fig. 4b, c). Given the discrepancies

between proxy-based temperature reconstructions for Siberia and computer-model simulations, the thermal gradient between

western Beringia and the subarctic NW Pacific may also differ. In the model simulation the thermal contrast between land and

ocean tends to become smaller since the negative temperature anomaly over western Beringia for the most part is more

pronounced than over the subarctic N-Pacific (Fig. 4b). This contrasts with the proxy compilation according to which the

thermal gradient was increased relative to present (Fig. 4c). As the model predicts a reduction of the thermal gradient the

preconditions for the increased landward air-flow are not given. In contrast a reduced thermal gradient would support a

northerly anomaly, which is in accordance with the simulated wind-patterns over Kamchatka (Fig. 4a). Hence, the

discrepancies between proxies and model-outputs concerning glacial summer temperature over western Beringia potentially

entail the mismatch regarding the atmospheric circulation patterns over the NW Pacific.

**5.2.3.    The deglaciation (18 ka-10 ka BP)**

The deglacial short-term variability strongly resembles the climate development in the N-Atlantic as $MAT_{ifs}$ follows the

deglacial oscillations recorded in the NGRIP-$\delta^{18}$O (Fig. 2b, i), particularly after 15 ka BP. $MAT_{ifs}$ clearly mirrors the

Bølling/Allerød (B/A)-interstadial, the Younger Dryas (YD)-cold reversal and the subsequent temperature increase into the

Preboreal (PB; Fig. 2b, i). This similarity suggests a strong coupling with climate change in the N-Atlantic realm and hence

variations in the AMOC-strength. The pronounced response to N-Atlantic climate change is in line with the temperature

development in the surrounding seas where the majority of climate-records shows a Greenland-like pattern (Ternois et al.,

2000; Seki et al., 2004; Max et al., 2012; Caissie et al., 2010; Praetorius and Mix, 2014; Meyer et al., submitted). This in-phase

variability is assumed to result from atmospheric teleconnections between the N-Atlantic and the N-Pacific Oceans (e.g.

Manabe and Stouffer, 1988; Mikolajewicz et al., 1997; Vellinga and Wood, 2002; Okumura et al., 2009; Chikamoto et al.,

2012; Max et al., 2012; Kuehn et al., 2014). While the effects of an atmospheric coupling with the N-Atlantic are undoubtedly

present between 15 and 10 ka BP their relevance is questionable during Heinrich Stadial 1 (HS1). The cold-spell between 18

ka BP and 14.6 ka BP as evident in the MAT$_{ifs}$ record may coincide with the HS1 in the N-Atlantic but initiates 2 ka earlier

than in NGRIP-$\delta^{18}$O. Considering that also SST records from the Western Bering Sea indicate that the Heinrich-equivalent

cold-spell commenced at approx. 16.5 ka BP (site 114KL, Meyer et al., submitted), the event in MAT$_{ifs}$ is probably not

associated with climate change in the N-Atlantic (Fig. 2b, g). This temporal offset cannot be explained by age-model

uncertainties in core 12KL since the error (1σ) of the calibrated radiocarbon ages is smaller than 100 yrs (Max et al., 2012). If

the cooling was not associated with climate change in the N-Atlantic, it could perhaps represents a local event on Kamchatka,

and potentially western Beringia, marking the abrupt end of the warm LGM-conditions. Since, to the knowledge of the authors,

such an event is not reported in the terrestrial realm of western Beringia, it is difficult to identify the driving processes. One

may speculate that the southerly flow abruptly weakened over Kamchatka.

A clear similarity between MAT$_{ifs}$ and NGRIP-$\delta^{18}$O establishes at approx. 15 ka BP. This has recently been described for the

SST in the marginal NW Pacific (Meyer et al., submitted) reconstructed for the same core site as investigated in the present

study (site 12KL, Fig. 2h). This record implies that the climate of the Kamchatka Peninsula until 15 ka BP was tied to the

climate change in the NW Pacific rather than to climate change in the Western Bering Sea (Fig. 2b, g, h, i). For SST this pattern

was explained by accumulation of AS waters in the NW Pacific, which likely overprinted the effect of the atmospheric

teleconnection by linking the western and the eastern basins of the N Pacific (Meyer et al., submitted). Hence, the effect of the

AS may have also determined temperature evolution on Kamchatka during the early deglaciation, restricting the teleconnection

to the period after 15 ka BP.

The clear and constant impact of N-Atlantic climate change between 15 and approx. 10 ka BP on Kamchatka is in agreement

with palynological data from the Kankaren Range/Northeast Siberia (Fig. 1a) where abrupt climatic changes corresponding to

the B/A and the YD are reported (Anderson and Lozhkin, 2015). Abrupt warming at the onset of the B/A is also evident in a

high resolution record from Lake Elikchan 4 (Lozhkin and Anderson, 1996; Kokorowski et al., 2008b) and may indicate a

linkage to N-Atlantic climate change. However, a climatic reversal equivalent to the YD is often absent in records from

northeast Siberia (east of 140°N and north of 65°N; Fig. 1a; e.g. Lake Jack London, Lake El'Gygytgyn and Wrangel Island;

Lozhkin et al., 1993, 2001, 2007; Lozhkin and Anderson, 1996; Nowaczyk et al., 2002; Nolan et al., 2003, Kokorowski et al.,



2008a,b), as compiled by Kokorowski et al. (2008a). By contrast, palynological data from Siberia (e.g. Lakes Dolgoe, Smorodynovoye and Ulkhan Chabyda, Fig. 1a) indicates that a YD climatic reversal was present west of 140°N (Pisaric et al., 2001; Anderson et al., 2002, Kokorowski et al., 2008a). This east-west gradient was explained by a westward shift of the East Asian Trough (EAT; normally situated over the central Beringian coast; Mock et al., 1998) which caused cooling west of 140°N by enhancing cold northerly winds, and together with an anticyclone over the Beaufort Sea brought warming through stronger easterlies into the region east of 140°N (Kokorowski et al., 2008a). The presence of a YD-cold reversal on Kamchatka and in the Kankaren Range implies that the southeastern edge of Siberia was probably not affected by the shifting EAT. Several general circulation models investigating the nature of teleconnections between the N-Atlantic and N-Pacific realms suggest that the westerly Jet played an important role by acting as heat-conveyor between the N-Atlantic and the N-Pacific-Oceans (e.g. Manabe and Stouffer, 1988; Okumura et al., 2009). Considering the modern average position of the westerly Jet (between 30 and 60°N) Kamchatka likely received the YD-cold reversal through the westerlies. Also, relatively strong marine influences from the N-Pacific may have induced cooling on Kamchatka and may have also affected the Kankaren Range. Together with the atmospheric patterns suggested by Kokorowski et al. (2008a), northward decreasing influences of the westerly Jet and the Pacific Ocean north may explain north-south differences in northeast Siberia.

### 5.2.4. The Holocene

Although not quite pronounced in magnitude, the long-term $MAT_{ifs}$ evolution during the Holocene is characterized by a mid-Holocene Thermal Maximum (HTM) between 8 and 4 ka BP which is followed by neoglacial cooling (Fig. 2b). This long-term development is in good agreement with existing climate records from central and southern Kamchatka (Fig. 2j) where pollen-based records indicate warm and wet conditions between 8 and 5 ka BP, which are associated with the HTM (Dirksen et al., 2013). According to $MAT_{ifs}$ the climate deterioration after the HTM started at approx. 4 ka BP. This timing is consistent with diatom-based climate reconstructions as well as chironomid-based temperatures from central and south Kamchatka (Dirksen et al., 2013; Hoff et al., 2014) and with re-advancing mountain glaciers (Savoskul et al., 1999, Barr and Solomina, 2014). As already discussed in previous studies this long-term temperature development is thought to respond to changes in mean summer insolation (Fig. 2b, e, j). As summarized by Brooks et al. (2015), the timing of the HTM (approx.



9-4 ka BP) on Kamchatka as well as in southern parts of eastern Siberia is delayed compared to northern parts of Chukotka

and Siberia where the HTM initiated at 9-8 ka BP (Biskaborn et al., 2012 and references therein; Nazarova et al., 2013b;

Anderson and Lozhkin, 2015). Since a similar delay of the HTM has been found in northern Europe (Seppä et al., 2009),

Brooks et al. (2015) concluded that the climate on Kamchatka was connected with the N-Atlantic realm by an atmospheric

coupling. Furthermore, the fact that Andrén et al. (2015) detected an 8.2 cooling-event in pollen-based climate records from

Kamchatka also points to a linkage with N-Atlantic climate.

Hence, it seems that the atmospheric linkage that determined climate variability during the deglaciation likely persisted into

the Holocene where it acted as an important driver for long-term climate changes as well as for abrupt, short-lived climatic

events.

## 6.   Summary and Conclusion

Based on the CBT/MBT'-paleothermometre a continuous LGM-to-late Holocene record of summer-temperature in Kamchatka

is presented. The temperature evolution and the driving mechanisms were investigated. The record allows inferences for the

glacial atmospheric circulation patterns (i) and to describe how regional climate drivers (such as oceanic and atmospheric

circulation) as well as global and supra-regional drivers (including $CO_{2atm}$, summer insolation and N-Atlantic climate

variability) influenced the climate change on Kamchatka (ii). The findings can be summarized as follows:

(i)     LGM-summer temperatures were as high as at present. The warm summers likely result from a change in the regional

atmospheric circulation including a stronger-than-present southerly winds over Kamchatka as a result of a stronger-than-

present anticyclone over the subarctic NW Pacific. This was potentially driven by increased thermal gradients between

western Beringia and the N-Pacific Ocean. The temperature reconstruction as well as the inferences for atmospheric

circulation contrasts with model simulations, which predict widespread cooling over Siberia and Kamchatka, and a

weakening of the NPH over the NW Pacific together with a reduction of southerly winds over Kamchatka. These

discrepancies underline the need of further investigations of the LGM-climate in the NW Pacific realm using

environmental indicators and model simulations.

During the LGM the warming effect of the altered regional atmospheric circulation likely balanced the cooling-effects of lowered $CO_{2atm}$ and summer insolation.

(ii) Abrupt millennial-scale fluctuations characterize the deglacial temperature development and represent the most prominent temperature changes during the past 20 ka. A first abrupt cooling-event at 18 ka BP marks the end of the warm LGM conditions and is likely caused by regional climate change, the origin of which cannot be identified, yet. From 15 ka onwards the temperature variations are obviously linked to climate change in the N-Atlantic, presumably via rapid atmospheric teleconnections, as the B/A-interstadial and the YD cold reversal are clearly present. Regional

differences regarding the presence of a YD-cold reversal in Siberia are possibly related to the position of the westerly Jet. During the Holocene the atmospheric linkage with the N-Atlantic remained active and together with summer insolation is a primary driver for the long-term temperature development.

**Acknowledgments**

The study was part of a PhD project funded by the Helmholtz association through the President's Initiative and Networking

Fund and is supported by GLOMAR – Bremen International Graduate School for Marine Sciences. Core SO201-2-12KL was recovered during cruise SO201-2 which took place in 2009 within the frame of the German-Russian research project "KALMAR" – Kurile-Kamchatka and Aleutian Marginal Sea Island Arc Systems: Geodynamic and Climate Interaction in Space and Time". We thank the Master and the crew of R/V SONNE for their professional support during the cruise. Dirk Nürnberg is thanked for providing sample material. Alexander Weise is acknowledged for his assistance on the geochemical

sample preparation in the laboratories at the University of Bremen. The data obtained during this study are available online on the "Pangaea"-database (www.pangaea.de).

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

**Figure captions**

Figure 1. (A) Overview of Beringia and the N Pacific. Site SO201-2-12KL is marked by a red star. Circles represent sites mentioned in the text. Black arrows indicate the surface circulation patterns of the N Pacific (e.g. Stabeno and Reed, 1994). BLB = Bering Land Bridge, KR = Kankaren Range, R = River, EKC = East Kamchatka Current. P = Peninsula. L= Lake (B) Map of the Kamchatka Peninsula and its major orographic units. CKD = Central Kamchatka Depression.

Figure 2. a) Concentrations of ΣbrGDGT of core 12KL. b) CBT/MBT' derived MAT$_{ifs}$ from Kamchatka (this study). Black pins represent the age control points from core 12KL (based on radiocarbon dating of planktonic foraminifera, Max et al.,





2012). c) BIT-index values of core 12KL (Meyer et al., submitted). d) Titanium/Calcium ratios (Ti/Ca, XRF-scan core

12KL, Max et al., 2012). e) Mean July insolation at 65°N (Berger and Loutre, 1991). f) Atmospheric $CO_2$ concentration

(EPICA dome C, Monnin et al., 2001). g) SST development in the marginal NW Pacific (site 12KL, Meyer et al., submitted).

h) SST evolution in the western Bering Sea (site 114KL, Meyer et al., submitted). i) NGRIP-$\delta^{18}$O (NGRIP, 2004) represents

climate change in the N Atlantic. j) Pollen-based temperature reconstructions from the CKD (after Dierksen et al., 2013).

Grey-shaded bars mark the HS1 and YD stadials.

Figure 3. Fractional abundances of all nine brGDGT in core 12KL, given in percentage relative to the amount of ΣbrGDGTs.

Figure 4. Comparison of proxy- and model-based inferences regarding glacial anomalies in temperature and atmospheric

circulation over the N Pacific and Beringia relative to present. (A) COSMOS-simulation for the SLP-anomaly over Beringia

and the N Pacific during the LGM (21 ka) relative to PI. Arrows represent the wind anomaly. Note that the model predicts a

northerly anomaly over Kamchatka. (B) COSMOS-simulation for the SAT-anomaly together with the wind-anomaly. (C)

Compilation of proxy based anomalies of summer air temperature in Beringia and of summer/autumn SST reconstructions in

the N Pacific for the LGM. Sites and corresponding references are given in the appendix, Table A1. Doted arrows sketch the

general summer anticyclone over the N Pacific, the NPH. Based $MAT_{ifs}$, the NPH and associated southerly winds over the

subarctic NW Pacific were stronger than at present (represented by solid arrow).

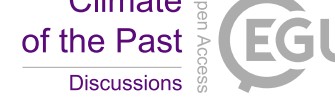

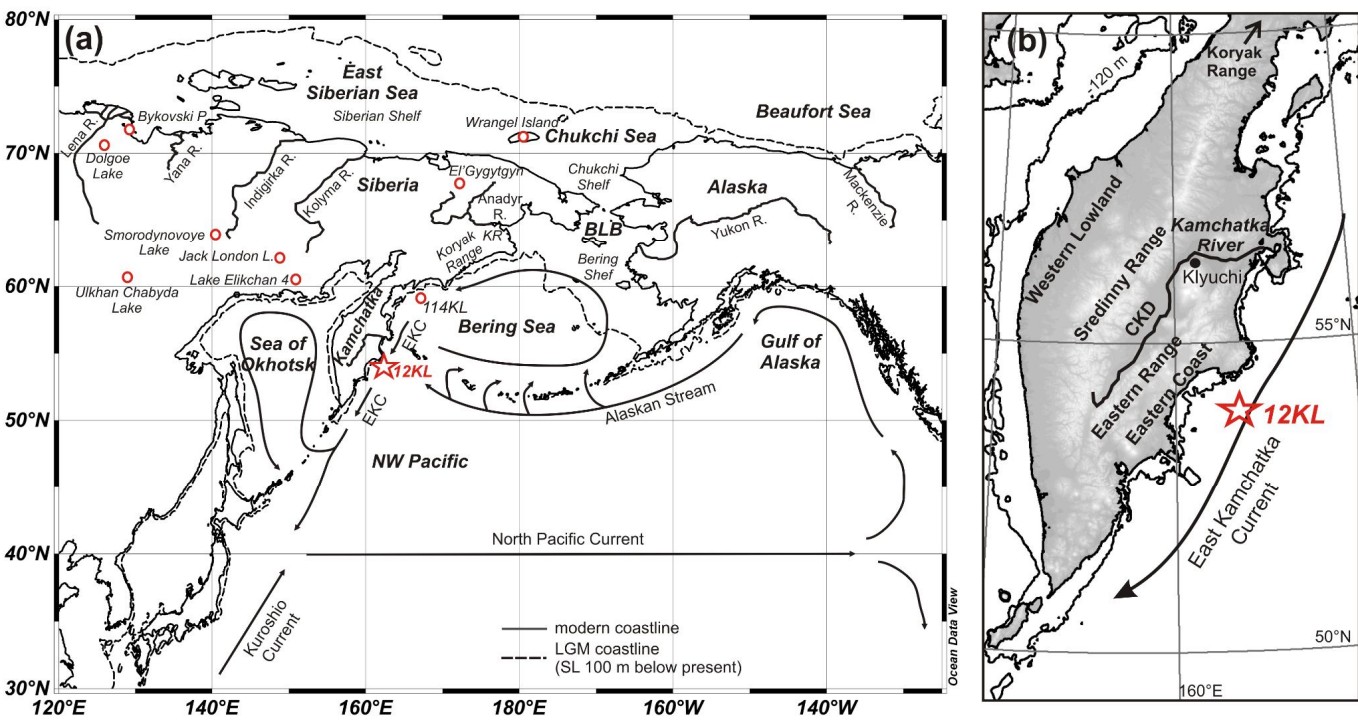

**Fig. 1**





Fig. 2





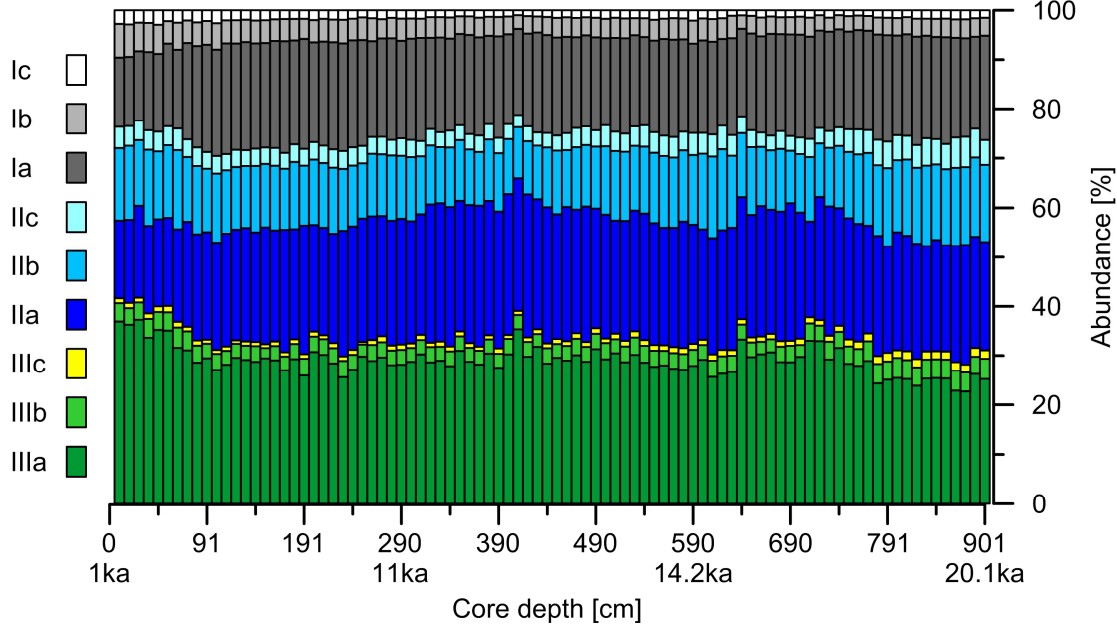

Fig. 3





**Fig. 4**



**Appendix A**

Table A1. Sites and references for the data compiled in Fig. 4c.

| No. | Site | Region | Proxy | Reference |
|---|---|---|---|---|
| 1 | SO201-2-12KL | NW Pacific/Kamchatka | CBT/MBT' | This study |
| 2 | SO201-2-12KL | NW Pacific | $TEX^L_{86}$ | Meyer et al., submitted |
| 3 | SO201-2-114KL | Western Bering Sea | $TEX^L_{86}$ | Meyer et al., submitted |
| 4 | MR0604-PC7 | Sea of Okhotsk | $U^{K'}_{37}$ | Seki et al., 2009, 2014 |
| 5 | XP98-PC2 | Sea of Okhotsk | $U^{K'}_{37}$ | Seki et al., 2004 |
| 6 | XP98-PC4 | Sea of Okhotsk | $U^{K'}_{37}$ | Seki et al., 2004 |
| 7 | MR00K03-PC04 | Sea of Okhotsk | $U^{K'}_{37}$ | Harada et al., 2004, 2012 |
| 8 | unknown | Sosednee Lake/Siberia | pollen | Lozhkin et al., 1993 |
| 9 | unknown | Oymyakon Depression/Siberia | beetle | Berman et al. (2011) |
| 10 | unknown | Middle stream of Indigirka River/Siberia | beetle | Berman et al. (2011) |
| 11 | unknown | Lower and middle reaches Kolyma River/Siberia | beetle | Berman et al. (2011) |
| 12 | Mkh | Bykovski Peninsula/Siberia | pollen/beetle | Kienast et al. (2005); Sher et al. (2005) |
| 13 | YA02-Tums1 | Yana lowlands/Siberia | pollen | Pitul'ko et al. (2007) |





| 14 | unknown | Indigirka Lowland/Siberia | beetle | Alfimov and Berman, (2001); Kieselev (1981) |
|----|---------|---------------------------|--------|---------------------------------------------|
| 15 | unknown | Kolyma Lowland/Siberia | beetle | Alfimov and Berman, (2001); Kieselev (1981) |
| 16 | unknown | Ayon Island/Siberia | beetle | Alfimov and Berman, (2001); Kieselev (1981) |
| 17 | PG1351 | Lake El'Gygytgyn | pollen | Lozhkin et al. (2007) |
| 18 | unknown | Markovo/Siberia | beetle | Alfimov and Berman, (2001); Kieselev (1981) |
| 19 | unknown | Anadyr River middle stream /Siberia | beetle | Berman et al. (2011); |
| 20 | Bering Shelf 78-15 | Shelf off Seward Peninsula/BLB | beetle | Elias et al. (1996, 1997); Elias (2001) |
| 21 | Zagoskin Lake | western Alaska | chironomids | Kurek et al. (2009) |
| 22 | Bering Land Bridge Park | Seward Peninsula/Alaska | beetle | Elias et al. (2001) |
| 23 | Burial Lake | St. Michael Island /BLB, Alaska | chironomids | Kurek et al. (2009) |
| 24 | Bluefish | Bluefish Basin/Alaska | beetle | Mathews and Telka, (1997); Elias et al. (2001) |
| 25 | SO202-27-6 | Gulf of Alaska | $U^{K'}_{37}$ | Maier et al. (2015) |



| 26 | PAR87A-10 | Gulf of Alaska | dinocysts | deVernal and Pedersen (1997) |
| 27 | MR97-02 St. 8s | NW Pacific | $U^{K'}_{37}$ | Harada et al. (2004, 2012) |
| 28 | MR98-05 St. 5 | NW Pacific | $U^{K'}_{37}$ | Harada et al. (2004, 2012) |
| 29 | MR98-05 St. 6 | NW Pacific | $U^{K'}_{37}$ | Harada et al. (2004, 2012) |
| 30 | unknown | Chaun Depression/Siberia | beetle | Berman et al. (2011) |