# Peer review of "Summer-temperature evolution on the Kamchatka Peninsula, Russian Far East, during the past 20,000 years"

_Climate of the Past, 2016_

## Referee Comment (RC1) · Anonymous Referee #1 · 17 Jun 2016

"Summer-temperature evolution on the Kamchatka Peninsula, Russian Far East, during the past 20,000 years" by V. Meyer et al. discusses a 20-kyr-long terrestrial temperature reconstruction for a region with relatively sparse continuous paleoclimate reconstructions. The methods used are at the forefront of climate reconstructions. The authors compile existing temperature reconstructions for the last glacial maximum from the region and compare these reconstructions with a climate model simulation. There are several aspects of the study, particularly related to the data-model comparison, that would need clarification and revision prior to publication in Climate of the Past. Clarifying these points will make this study a strong and robust contribution to Beringian paleoclimate reconstruction.

Specific comments:

Line 95: list the number of samples analyzed and the approximate or average depth and time sampling resolution

Line 133-134: How exactly what the standard deviation measured? Is this for this lab or labs in general? Was a standard measured regularly among sample injections, or were the repeat measurements of the samples? What is the pooled standard deviation of samples that were run multiple times (if any?...and if none, then that is important to report)? Please clarify in the text.

Line 136: it is critical to report the calibration error, which is much larger than the analytical error. While relative changes in a single record are likely real, the absolute temperature change is difficult to pinpoint because of this large calibration error. Please clarify in the text.

Line 148: does the word "glacial" belong here? it doesn't make sense. Please clarify in the text.

Line 148-149: what parameters were used in this simulation? While the reader can check the references given here, it would be good to briefly summarize the parameters that were used to force the model and the parameters that were changed between the glacial and preindustrial runs (ice volume? sea level? orbital forcing? greenhouse gases? land cover? etc….) Please clarify in the text. Ice volume, sea level, and land cover are of particular importance for this region, where a large amount of land was exposed during the LGM.

Line 157: what does "integrated" mean? is this the model spin up? was this done twice, once for each the LGM and PreIndustrial runs? Please clarify in the text.

Line 181: Because this record is not in the North Atlantic, it would be best to avoid using terms that are related to North Atlantic climate change (ie. Bolling Allerod) in this results-oriented portion of the paper. When the authors later discuss links with the

[Figure]

North Atlantic, these North-Atlantic-based terms can and should be introduced.

Line 204: It seems to me that the change from exposed land during the LGM to ocean during the PreIndustrial run over the land bridge would be a source of large changes in modeled SAT. This aspect is important to address, not only how this is handled in the model (is this exposed land in the LGM simulation?), but also how this could affect SAT in the model, and whether that is similar to the real-world effects. I would question whether these anomalies are even meaningful, and would need more explanation of what the changes mean, because of the changes from land to ocean surface.

Line 227: using slashes to indicate opposite effects is confusing. I suggest removing them and adding a phrase at the end of the sentence, like "with the opposite effect occurring with low terrigenous input". See [Robock, 2010] for a humorous take on how confusing it can be to use slashes to express opposites.

Line 231: do the authors mean "in marine areas where brGDGTs are thought..."? please clarify.

Lines 242 and 244: what are the uncertainties or standard deviations on these temperature observations? please clarify.

Line 251: cite PMIP?

Line 267: clarify whether this attribution was by previous studies, or by this study.

Line 280 and others: Clarify in the text what proxy was used to produce this Sea of Okhotsk SST reconstruction.

Line 285: 1°C is well within calibration error of these proxies, and is important to mention in the text.

Line 325-328: It seems as if the final two sentences in this paragraph say opposite things. Can this be clarified?

Line 330: How robust or meaningful is this warmer-than-present temperature, given

that there were large changes in surface conditions (land to ocean) from the LGM to present? I would expect summer temperatures to be quite warm over land, as dark soils can retain quite a bit of heat, whereas sea water remains much cooler. It is important to address the changing surface conditions in the text.

Line 344: It might clarify to add the following wording: "potentially explain the mismatch between model and proxy..."

Line 351: does the term "in the surrounding seas" refer to the Pacific or the Atlantic? It is unclear, as both are mentioned in this section. Please clarify.

Line 360: in addition to the age model error, the authors must also discuss error in marine reservoir corrections, and it would be helpful show age uncertainties in Fig. 2 time series.

Line 367: clarify what proxy is used to reconstruct SST in the NW Pacific.

Line 368-369: this sentence is unclear, please rewrite and clarify.

Line 370: What does AS stand for? Perhaps just spell out the full term.

Line 366-373: why would ocean waters place a "restriction" on atmospheric teleconnections? Can this be clarified?

Line 405: I don't understand how the HTM is delayed on Kamchatka relative to other parts of Siberia, as both have the same beginning time (9ka). Can this point either be deleted or made more clear?

Lines 407-410: these speculative connections with the North Atlantic seem like a stretch and could be explained by other, regional climate forcing mechanisms. Perhaps it would be best to remove these sentences?

Lines 428-429: It is unclear what this sentence means. Please clarify.

Fig. 2: Plot age and proxy uncertainty envelopes for all data from core 12KL. Proxy

uncertainty includes analytical uncertainty (relatively small) and calibration uncertainty (quite large, relative to the signal). This is important to report.

Fig. 4: show the model LGM land boundaries and the PI land boundaries. Are these annual or summer anomalies? Clarify in the figure caption.

Technical corrections: Line 12: Branched Glycerol. . .does not need to be capitalized

Line 35: clarify what "next to" means: rather than? or in addition to?

Line 53 (and elsewhere in the text): I think the authors mean 150°W here, and this same typo is made elsewhere (e.g., line 385).

Line 74: clarify what "over the northern shelves of central Beringia" means. Is this a geographic location? Could this be highlighted on a map or described in more clear terms?

Line 175: it might help to add the word "respectively" to the end of the sentence that lists the percentages.

Line 184: change "with approx." to "at approx."

Line 194: Add "North" before "American continent"

Line 203: remove the w in "now"

Line 209: it might be more clear to say that the SAT anomaly becomes stronger or becomes more pronounces from east to west (because the anomaly is actually decreasing from east to west).

Line 226: this is an incomplete sentence.

Line 248: add comma between climate and according

Line 250: change 'computer' to 'general circulation'

Line 253: change "ice caps" to "ice sheets"

Line 255: is CO2atm defined prior to this? If not, then define here.

Line 261: add "summer" between present and conditions.

Line 412: define what "it" refers to.

Line 420: change "were as high as at present" to "were similar to present temperatures" or something to that effect.

Line 421: remove "a" before "stronger-than-present"

Robock, A. (2010), Parentheses Are (Are Not) for References and Clarification (Saving Space), Eos, Transactions American Geophysical Union, 91(45), 419, doi:10.1029/2010EO450004.

---

## Referee Comment (RC2) · Anonymous Referee #2 · 20 Jun 2016

The manuscript "Summer-temperature evolution on the Kamchatka Peninsula, Russian Far East, during the past 20,000 years" presents a terrestrial MAT temperature reconstruction based on branched glycerol dialkyl glycerol tetraethers analyzed in a radiocarbon-dated marine sediment core from the NW Pacific margin. The interpretation of the GDGTbr records as reflecting terrestrial temperatures is well defended, and independently-dated records of this nature are sorely needed for the NW Pacific. The paper would be improved by a more rigorous handling of some of the more quantitative aspects of comparison with other regional/global records: specifically improved evaluation/presentation of the errors in the temperature reconstruction, as well as the chronology of the core. A more quantitative treatment of the comparison of the Kam-

chatkan MAT temperature record to the Greenland ice-core NGRIP d18O record would also strengthen the paper. This is particularly important given the strong statements the authors make w/re the North Atlantic 'obviously' driving NW Pacific climate. Detailed suggestions are below.

Abstract Line 12: Rather than 'western continental margin off Kamchatka/marginal Northwest Pacific ' suggest 'western continental margin off Kamchatka in the Northwest Pacific'

1. Introduction Line 32-37: Long, awkward sentence. Suggest reworking for clarity.

Line 37: Unsure why this sentence begins with 'Particularly' in context of previous sentence.

On line 40: Statement that majority of sea surface temperature records from the sub-arctic NW Pacific and marginal seas mirror N. Atlantic climate oscillations needs to be qualified. I am assuming that this claim pertains to millennial scale N. Atlantic climate oscillations as recorded from ice core d18O? If so, Caissie et al. 2010 reference is for a surface ocean temperature record of multi-millennial scale resolution with very limited chronological control that bears little resemblance in structure to NGRIP d18O outside of a crude transition from apparently full glacial to interglacial conditions between 12-11 ka. Of the 6 records from the 'NW Pacific and marginal seas' presented by Max et al., 2010, while the nearly all the color b* records resemble deglacial NGRIP d18O in structure, only one of the attendant SST records (the NW Pacific core SO201-12-KL that is also the subject of this paper) looks anything like NGRIP d18O at millennial scales. I can't comment on the Meyer et al. reference as it's unpublished.

If correct, this rather sweeping assertion that NW Pacific SST mirrors N. Atlantic climate has fairly important implications vis-a-vis the following suggestion that N Atlantic teleconnections control deglacial temperature development in the N. Pacific. However this assertion seems poorly defended by the references offered in the text at this point and at the very least needs further qualification/elaboration.

Line 44: In the marine environment you only address records from the broader NW Pacific, but then for terrestrial records you include records from the Alaskan portion of Beringia. Why is there no discussion of the well-dated marine records of climate from the NE Pacific, or alternatively, why are the terrestrial Alaskan records being included in this discussion?

2. Regional Setting Line 71: Suggest replacing 'which are' with a comma.

Line 75: Why is Jet capitalized? Should it be 'westerly jet stream'? Similarly 'Jetstream' in line 78 should also be two lowercase words.

Line 82: Add comma after 'ranges'.

Line 84: Would read better as 'Mean temperatures averaged for the entire Peninsula range from...'. Alternatively, place a comma after 'Peninsula'.

3. Materials and Methods 3.1 Core material and chronology

Although you reference Max et al. (2012) for details of chronology, as it's highly pertinent to this paper it would be nice to have some basic information offered on the core length and the number of radiocarbon dates that constrain accumulation. Similarly, as these results are the subject of another paper, a mention of the mean sedimentation rates in the Holocene and deglacial sections of the core (properly cited) would be useful to the reader.

3.2 Lipid Extraction Line 101: No need for a comma after n-hexane.

3.4 Temperature determination

I'm unclear from this how the BIT-index controls for GDGT's from fresh water environments?

In this and previous (3.3) section, in many locations in the text I'm unclear on what precipitates the use of the abbreviation GDGT as opposed to brGDGT? I would have guessed it was branched versus all Glycerol Dialyl Glycerol Tetraethers, but in some

cases (if I'm not mistaken) GDGT appears to be used interchangeably with brGDGT. Really this comment could extend to entire manuscript.

Line 145: pluralize 'sample'. Also no need for 'present'.

4. Results 4.1 Concentrations and fractional abundance of brGDGT

It would seem to me that Figure 3 (and the discussion of it) should precede Figure 2 in the text.

4.2 Temperature development over the past 20 ka Line 178: This is a very narrow definition of the late Holocene (1 ka BP), and it only affords you one data point in the record of 12KL to compare to the dozen or so available for the 2 ka window afforded the glacial (18-20 ka BP). Perhaps consider broadening the definition of 'late Holocene' and when presenting a surface temperature include a standard deviation that encompasses both analytical/calibration error as well as observed variability over that time interval. Also, here and every future instance, why is approximately abbreviated?

While I don't doubt that the glacial temperatures are statistically 'the same' as those in the late Holocene, this could easily be presented quantitatively.

Line 179: In this and every instance throughout the paper, when giving a temperature, present that value in the context of its uncertainty.

Line 179-182: This pair of sentences is awkward and read poorly. The discussion of the single warm data point at 16 ka reads like a stream of consciousness as opposed to a well-digested scientific observation.

Line 181: If you're going to present ages down to the century scale, you need to include estimates of temporal error that reflect the chronological control of the core.

Line 187: What is the average mid-Holocene thermal maximum temperature between 8.0-4.0 ka BP (with errors)? Perhaps present the average temperature in that window, then give the highest temperature reached and the age (again, with errors) that that

peak temperature is observed.

Line 188: How do you determine when the cooling trend is initiated? It would seem that the cooling arguably begins closer to 5 ka, but then again if you're interpreting at this level (and you probably shouldn't) you could argue the cooling stops by 3 ka.

Line 189: This last sentence needs to be quantitative. Also, when calculating variance, remember to use equivalent temporal windows for the Holocene and deglacial and smooth the record to a constant resolution.

5. Discussion

5.1 Sources of brGDGT and implications for CBT/MBT'-derived temperatures Line 219: Either here or in the methods section some very basic discussion of how to interpret the BIT-values should be given.

Line 221: Eminent might not be the right word choice for this sentence. Perhaps 'Marine settings where terrigenous input is low are particularly sensitive to bias from in-situ production, thus non-soil derived brGDGTs potentially have a considerable effect on the temperature...'

Line 225: Again, would suggest minor reworking. Perhaps 'Ti/Ca-ratios reflect the proportion of terrigenous and marine derived inorganic components of the sediment, and can be used as an estimator of terrigenous input'.

Line 226: 'With relatively high values at 15.5 and 12 ka BP, and minima at 14 and 11 ka BP' is an incomplete sentence. Also, again, if presenting chronologies at the centennial scale really need to give errors on those ages.

Line 244: 'Mai' should be 'May'.

5.2 Temperature evolution over the past 20 ka 5.2.1 The LGM (20-18 ka) – warm summers and the regional context

What definition of LGM are you using? Should give a reference. Clark et al., 2009 is

the most widely used citation that I'm aware of and they define global LGM as ending at 19 ka.

Line 251: While you could say there was a 'cooling tendency' from MIS-3 into the LGM, since time moves forward when comparing the LGM to the Holocene it would be better to say 'Generally cooler LGM temperatures are thought to result from...'

Line 257: What does 'BLB' stand for?

Line 260: No need to hyphenate 'insect-data'. Also suggest rewording to 'Markovo, and ElGygytgyn and Jack London lakes'

5.2.2. Controls on MATifs

In this section you identify a possible seasonal bias in alkenone-based SST reconstructions towards warmer temperatures and dismiss them in favor of TEXL86 reconstructions. You then discuss the results of the TEXL86 reconstruction for site 12KL currently submitted for review. However there is no discussion of the already-published alkenone-based SST record for 12KL of Max et al., (2012), nor is there a presentation of this record alongside the TEXL86 record from the same site in Figure 2. For the period of overlap, it would appear that at least at this location the alkenone SST's are several degrees colder than the TEXL86 temperature reconstruction. Why would this be?

Line 266: Need to clarify that you're discussing warm Siberian summers during LGM

Line 288: As the paper Meyer et al., (submitted) has yet to pass through peer review, probably best to state that the relatively warm SST's at site 12KL may be explained by stronger-than-present influence of the Alaskan Stream.

5.2.3 The deglaciation (18 ka-10 ka BP)

Define/defend the use of the words 'strong' and 'clear' when describing the resemblance between the N-Atlantic d18O and 12KL MAT. Can you calculate covariance

between the normalized/equivalently smoothed NGRIP d18O and 12KL MAT? To my eye they appear quite different: the Y-D is greatly compressed in the Kamchatka MAT record, the trend from the LGM to HS1 in 12KL is completely absent in NGRIP. A climate oscillation in HS1 apparently comparable in magnitude and duration to the regional expression of the Y-D (although a warming as opposed to cooling event) at 16 ka with no analogue in NGRIP is discarded from interpretation. I'm not arguing that there are similarities, but to say it's obvious or 'undoubtable' that the North Atlantic is driving NW Pacific climate via atmospheric teleconnection is a strong claim that needs to be quantitatively defensible. If this can't be done in the context of this paper, perhaps dial the tone of the text down a bit.

Also, as stated earlier in the text, when comparing 12KL to NGRIP at centennial scales chronological uncertainties in 12KL need to be addressed and stated.

5.2.4 The Holocene

The statement at line 411: "Hence it seems that the atmospheric linkage (with the N-Atlantic) that determined climate variability during the deglaciation likely persisted into the Holocene where it acted as an important driver for long-term climate changes as well as abrupt, short-lived climate events." seems poorly defended by the visual similarity between NGRIP d18O and Kamchatka MAT in Figure 2. To my eye the Holocene in the MAT record appears more variable, while the mid-Holocene thermal maximum and neoglacial cooling described for the NW Pacific region are absent in NGRIP. Quantitatively evaluating the covariance between these records would be challenging at best as the current chronology for 12KL is virtually unconstrained in the Holocene.

If this statement remains in the discussion/conclusions, at the very least some discussion of what is meant by 'long-term climate changes' versus 'abrupt, short-lived climatic events'.

6. Summary and Conclusions

Line 415-419: This introduction to the conclusions reads awkwardly.

Line 420: Perhaps replace 'likely' with 'may' or 'could' as there is no evaluation of statistical certainty of this hypothesis.

Line 433: Again, the use of the word 'obvious' to describe the role of N-Atlantic climate in driving the NW Pacific seems somewhere between bombastic and unfounded. There are some similarities in deglacial climate, there are differences, and as yet these remain poorly quantified in the manuscript.

Figures

Figure 1: Could some kind of shading be used to more clearly denote Holocene land-masses? With apparently identical solid lines used to denote boundaries of continents, ocean currents, and rivers it's a bit difficult to visually parse.

Figure 2: As this figure includes the TEXL86 SST record from Site 12KL to be published in Meyer et al., submitted, it should probably also include the deglacial alkenone SST record from site 12KL published in Max et al., 2012.

Figure 3: As mentioned in my comments on the results section, I think this figure should be reversed with Figure 2 in its presentation order in the text. Also, instead of giving ages at 4 depths in the core, could a secondary axis with appropriately dilated/compressed ticks be added for age alongside the depth scale? If this isn't possible, would almost suggest it would be better to present results versus time than versus depth to facilitate comparison to Figure 2.

---

## Author Comment (AC1) · 4 Aug 2016

Dear reviewer, thank you very much for your detailed and helpful review on our manuscript. Please, see below how we intend to address your comments and suggestions to revise our manuscript..

Specific comments Line 95: list the number of samples analyzed and the approximate or average depth and time sampling resolution

We will add to line 95: "For this study we used the same samples as Meyer et al, (2016). The core was sampled in 10 cm steps providing an average temporal resolution of approximately 200 years.

Line 133-134: How exactly was the standard deviation measured? Is this for this lab or labs in general? Was a standard measured regularly among sample injections, or were the repeat measurements of the samples? What is the pooled standard deviation of samples that were run multiple times (if any?...and if none, then that is important to report)? Please clarify in the text.

The standard deviation derives from repeated measurements of a standard sediment extract (n=7) which had been treated in the same way as the samples. So, the standard deviation is for the sample preparation and measurement processes in our lab. Repeated measurement for the samples of core 12KL were not possible, because of small amounts. We will add the following sentence to line 133-134: "From repeated measurements of a lab-internal standard sediment extract (n=7) the standard deviations for CBT and MBT' were determined as 0.01 and 0.04, respectively."

Line 136: it is critical to report the calibration error, which is much larger than the analytical error. While relative changes in a single record are likely real, the absolute temperature change is difficult to pinpoint because of this large calibration error. Please clarify in the text. In the results and discussion sections we will refer to the calibration error whenever absolute temperature values are given using xy $\pm$ 5°C.

Line 148: does the word "glacial" belong here? it doesn't make sense. Please clarify in the text.

No, the word does not belong here. It will be deleted. Thanks!

Line 148-149: what parameters were used in this simulation? While the reader can check the references given here, it would be good to briefly summarize the parameters that were used to force the model and the parameters that were changed between the glacial and preindustrial runs (ice volume? sea level? orbital forcing? Greenhouse gases? land cover? etc: : :.) Please clarify in the text. Ice volume, sea level, and land cover are of particular importance for this region, where a large amount of land was exposed during the LGM.

We will append the following paragraph to line 156. "External forcing and boundary conditions are imposed according to the protocol of PMIP3 for the LGM (available at http://pmip3.lsce.ipsl.fr/). The respective boundary conditions for the LGM comprise orbital forcing, greenhouse gas concentrations ($CO_2$=185ppm; $N_2O$=200ppb; $CH_4$=350ppb), ocean bathymetry, land surface topography, run-off routes according to PMIP3 ice sheet reconstruction and increased global salinity (+ 1 psu compared to modern value) to account for a sea-level drop of ∼116 m. The glacial ocean was generated through an ocean-only phase of 3000 years and coupled phase of 3000 years (LGMW in Zhang et al., 2013). The land cover is calculated interactively in the climate model which has an interactive land surface scheme and vegetation module (Brovkin et al. 2009). The modular land surface scheme JSBACH (Raddatz et al., 2007) with vegetation dynamics (Brovkin et al., 2009) is embedded in the ECHAM5 atmosphere model. The background soil characteristics (which are described in Staerz et al., 2016) are set to the values which are closest to the pre-industrial land points."

Line 157: what does "integrated" mean? is this the model spin up? was this done twice, once for each the LGM and Pre-Industrial runs? Please clarify in the text.

Integration means "simulated years". In order to clarify we will write: "For both, PI and LGM conditions, the climate model was integrated twice for 3000 model years and provides monthly output (Wei et al., 2012; Wei and Lohmann, 2012; Zhang et al., 2013)."

Line 181: Because this record is not in the North Atlantic, it would be best to avoid using terms that are related to North Atlantic climate change (ie. Bolling Allerod) in this results-oriented portion of the paper. When the authors later discuss links with the North Atlantic, these North-Atlantic-based terms can and should be introduced.

This is a good point, thanks. We will replace the Atlantic-related terms by dates. For example: "...until the beginning of the Bølling/Allerød" will turn into:"...until 14.6 ka BP."

Line 204: It seems to me that the change from exposed land during the LGM to ocean during the PreIndustrial run over the land bridge would be a source of large changes in modeled SAT. This aspect is important to address, not only how this is handled in the model (is this exposed land in the LGM simulation?), but also how this could affect SAT in the model, and whether that is similar to the real-world effects. I would question whether these anomalies are even meaningful, and would need more explanation of what the changes mean, because of the changes from land to ocean surface.

As already mentioned in the reply to the comment on line 148-149, we will add a paragraph describing the model setup in more detail. The effects of the exposed land in the model and the proxy world will be addressed in the discussion (see comment on line 330).

Line 227: using slashes to indicate opposite effects is confusing. I suggest removing them and adding a phrase at the end of the sentence, like "with the opposite effect occurring with low terrigenous input". See [Robock, 2010] for a humorous take on how confusing it can be to use slashes to express opposites.

We suggest to replace the sentence from line 227 by: "A similar pattern is visible in $\Sigma$brGDGT-concentrations as these increase during intervals of enhanced terrigenous input (high Ti/Ca-values) and decrease when terrigenous input is relatively low (low Ti/Ca values, see Fig. 2b, d). This suggests that brGDGTs are terrigenous."

Line 231: do the authors mean "in marine areas where brGDGTs are thought..."? Please clarify.

Yes, we do. In order to clarify we will insert "marine" before "areas".

Lines 242 and 244: what are the uncertainties or standard deviations on these temperature observations? Please clarify.

Please, see comment on line 136.

Line 251: cite PMIP?

Good idea. We will mention PIMP and cite Braconnot et al. (2012).

Line 267: clarify whether this attribution was by previous studies, or by this study.

In order to clarify we suggest the following modification: "In previous studies the warm Siberian summers during the LGM were attributed to increased continentality, which would arise from the exposure of the extensive Siberian and Chukchi shelves at times of lowered sea-level (Fig. 1a; e.g. Guthrie, 2001; Kienast et al., 2005; Berman et al., 2011)."

Line 280 and others: Clarify in the text what proxy was used to produce this Sea of Okhotsk SST reconstruction.

The sentence encompassing lines 279 and 280 begins with: Alkenone-based SST... This implies the application of UK'37. "UK'37" could be used instead of "Alkenone-based" or can be put in parentheses, in order to specify. In all other instances where we did not mention the SST proxy (lines 285-286) it will be implemented. It is TEXL86 in every case.

Line 285: 1°C is well within calibration error of these proxies, and is important to mention in the text. The calibration error of the calibration applied to the TEXL86 at site 12KL is 1.7°C (Meyer et al., 2016 and references therein). We will write 1 ±1.7°C

Line 325-328: It seems as if the final two sentences in this paragraph say opposite things. Can this be clarified?

In order to clarify we will append an additional paragraph to line 328. The paragraph comprising lines 297-307 will be shifted behind the additional one. The resulting paragraph will be as follows: "Considering the consistency of different GCMs, the anticyclonic anomalies over North America as well as resulting cyclonic anomalies over the N-Pacific seem to be a robust feature of the glacial atmospheric circulation. Therefore, it is unlikely that the increased influence of the NPH over Kamchatka (as inferred from MATifs) was caused by a strengthening of the NPH. So, we hypothesize that the NPH

may have weakened in response to strong anticyclonic anomalies over the LIS, but at the same time shifted westward relative to today. Since the NPH is centered over the NE Pacific under present-day conditions a westward shift would automatically increase the strength of the southerly flow over the NW Pacific. This may explain why the influence of the NPH became stronger over the NW Pacific despite a general weakening of the anticyclone. Interestingly, the general patterns of temperature change over Beringia and the N Pacific Ocean (as inferred from the proxy compilation, Fig. 4c) suggests that the LGM thermal gradient between western/central Beringia and the N-Pacific was increased relative to today (Fig. 4c). While warm summers were widespread in western Beringia (Alfimov and Berman, 2001; Kienast, 2002; Kienast et al., 2005; Sher et al., 2005; Berman et al., 2011), the majority of SST records from the open N Pacific and the Bering Sea indicate colder conditions during the LGM (Fig. 4c; deVernal and Pedersen, 1997; Seki et al., 2009, 2014; Kiefer and Kienast, 2005; Harada et al., 2004; 2012; Maier et al., 2015; Praetorius and Mix, 2014; Praetorius et al., 2015; Meyer et al., 2016). Under the assumption that alkenone-based reconstructions of LGM SST in the Sea of Okhotsk (Seki et al., 2004, 2009; Harada et al., 2004, 2012) are biased, also the Sea of Okhotsk may have been significantly colder than at present as suggested by TEXL86-based SST reconstruction (Seki et al. 2009; 2014). An increased thermal gradient between the subarctic N Pacific and western Beringia would translate into an increased pressure gradient between the continental low-pressure over western Beringia and the high pressure over the subarctic NW Pacific. As this would intensify the southerly flow over Kamchatka relative to today, this mechanism may supported a westward displacement of the NPH."

Line 330: How robust or meaningful is this warmer-than-present temperature, given that there were large changes in surface conditions (land to ocean) from the LGM to present? I would expect summer temperatures to be quite warm over land, as dark soils can retain quite a bit of heat, whereas sea water remains much cooler. It is important to address the changing surface conditions in the text.

We included the land boundaries used in the PI and LGM simulations into Figure 4a and 4b (as requested in your comment on Figure 4). Based on these modifications we will also change the paragraph of lines 329-337 as follows: "The distribution of temperature anomalies in the COSMOS simulation shows a different pattern than the proxy compilation (Fig. 4b and c). The model predicts a widespread cooling over Siberia and Kamchatka where the majority of proxy data suggests warmer or equal temperatures relative to present. Relatively warm summers in western and central Beringia (as inferred from the proxy data) have been explained by increased continentality due to the exposure of the Siberian, Bering and Chukchi Shelfs during the LGM (Guthrie, 2001; Kienast et al., 2005; Berman et al., 2011). In the model the impact of continentality may be comparable to the proxy world over the eastern Siberian and the northern Chukchi Shelf since SAT anomalies are between -1 and +1°C (Fig. 4b) implying that summer SAT were similar to PI conditions. Also, positive anomalies over parts of the Bering and Chukchi Shelf are likely associated with the shelf exposure (Fig. 4b). However, for the latter, easterly to southeasterly wind anomalies over south Alaska and the BLB (Fig. 4b), may also play a role. Given the discrepancies between model and proxy based results for Siberian SAT it seems that the effect of continentality in the COSMOS simulation is weaker than in the proxy world and that other factors are more important. Reduced CO2atm is generally regarded a prominent cause for globally lowered temperatures during the LGM (e.g. Kageyama et al., 2006; Shakun et al., 2012). As a regional factor, cooling over the Arctic Ocean combined with northerly anomalies in the wind patterns over the East Siberian Sea may have enhanced the advection of cold arctic air masses to Siberia, a mechanism supporting SAT decrease in Siberia (Mock et al., 1998). Similarly, northerly anomalies are also present over Kamchakta and those are in agreement with summer cooling on the Peninsula (Mock et al., 1998). Given the discrepancies between proxy-based temperature reconstructions for Siberia and the ESM, the thermal gradient between western Beringia and the subarctic NW Pacific seems to differ, too. In the model simulation the thermal contrast between land and ocean tends to become smaller since the negative temperature anomaly over western

Beringia for the most part is more pronounced than over the subarctic N-Pacific (Fig. 4b). This contrasts with the proxy compilation according to which the thermal gradient was increased relative to present (Fig. 4c). As the model predicts a reduction of the thermal gradient the preconditions for the increased landward air-flow are not given. In contrast a reduced thermal gradient would support a northerly anomaly, which is in accordance with the simulated wind-patterns over Kamchatka (Fig. 4a). Hence, the discrepancies between proxies and model-outputs concerning glacial summer temperature over western Beringia potentially explain the mismatch between model and proxy based reconstructions of the atmospheric circulation patterns over the NW Pacific."

Line 344: It might clarify to add the following wording: "potentially explain the mismatch between model and proxy..."

You are right, the sentence reads better with your suggestion. We will adapt it (see previous comment).

Line 351: does the term "in the surrounding seas" refer to the Pacific or the Atlantic? It is unclear, as both are mentioned in this section. Please clarify.

It refers to the surrounding seas of Kamchatka, i.e. the Bering Sea, the subarctic NW Pacific and the Sea of Okhotsk. We intend to put "in the surrounding seas of Kamchatka (Bering Sea, NW Pacific, Sea of Okhotsk)" along this line.

Line 360: in addition to the age model error, the authors must also discuss error in marine reservoir corrections, and it would be helpful show age uncertainties in Fig. 2 time series.

This is an important point, thanks. We will implement a short paragraph describing the uncertainties introduced by the AMS dating as well as by the assumptions for reservoir ages in section 3.1.

Paragraph to be added in section 3.1: Max et al. (2012) converted radiocarbon ages into calibrated calendar ages using the calibration software Calib Rev 6.0 (Stuiver and

[Figure]

Reimer, 1993) with the Intcal09 atmospheric calibration curve (Reimer et al., 2009). A constant reservoir age of 900 years was assumed for the entire time-interval covered by the core (Max et al., 2012). The uncertainty of AMS dating was smaller than + - 100 years (Max et al., 2012). Another important issue are changes in reservoir ages of the surface ocean, in particular during the last deglaciation (Sarnthein et. al., 2015). However, recent studies suggest that reservoir ages of the Bering Sea and the N-Pacific varied by less than 200 years during the last deglaciation (Lund et al., 2011; Kühn et al., 2014) and are within the range of reservoir ages originally assumed by Max et al. (2012).

As for line 360, we will refer back to this paragraph: "As elaborated in section 3.1 the uncertainty of the age control is a few hundred years. Therefore, uncertainties in the age model are unlikely to explain the temporal offset."

Concerning age uncertainty estimates in Figure 2, we will not include any graphical features, e.g. bars, as those would make the Figure look very busy if we also include error bars for the temperature (as requested in your comment on Figure 2). The reader will be able to read details about the uncertainty in section 3.1 (see comment on line 360).

Line 367: clarify what proxy is used to reconstruct SST in the NW Pacific.

It is TEXL86. We will include this information.

Line 368-369: this sentence is unclear, please rewrite and clarify.

Indeed, this sentence was confusing. We will replace it by the following: "This has recently been described for the SST at this core site in the marginal NW Pacific, which was reconstructed using the TEXL86 proxy (Meyer et al., 2016)."

Line 370: What does AS stand for? Perhaps just spell out the full term.

Thanks, "AS" stands for "Alaskan Stream". The abbreviation will be defined here.

Line 366-373: why would ocean waters place a "restriction" on atmospheric teleconnections? Can this be clarified?

This has been elaborated in Meyer et al., 2016 on the basis of two SST records from the Western Bering Sea and the NW Pacific. It was concluded that the Alaskan Stream connected the deglacial SST development of the NW Pacific with the Gulf of Alaska, thereby causing different SST developments in the NW Pacific and the Bering Sea. Details can be found in Meyer et al., 2016 and references therein.

We will reword the sentence to: "the effect of the AS may have also determined temperature evolution on Kamchatka during the early deglaciation, which would explain why the linkage to the North Atlantic did not initiate before 15 ka BP." avoiding the confusing expression "restricting".

Line 405: I don't understand how the HTM is delayed on Kamchatka relative to other parts of Siberia, as both have the same beginning time (9ka). Can this point either be deleted or made more clear?

Thanks, there is a typo in the parentheses describing the beginning of the HTM in Siberia. This will be corrected. We will write: "As summarized by Brooks et al. (2015), the beginning of the HTM on Kamchatka (approximately 9-8 ka BP; see Brooks et al., 2015 and references therein) on Kamchatka as well as in the sub-Arctic part of eastern Siberia (Nazarova et al., 2013b) is delayed compared to regions further north towards the Arctic Ocean where the HTM initiated at 10 ka BP (Biskaborn et al., 2012 and references therein)."

Lines 407-410: these speculative connections with the North Atlantic seem like a stretch and could be explained by other, regional climate forcing mechanisms. Perhaps it would be best to remove these sentences?

We will delete the sentences in lines 409-413 as the presence of an 8.2 event on Kamchatka is debatable on the basis of the existing records and it is not apparent in

[Figure]

MATifs. We leave the sentences in line 407 and 408 in as the timing of the HTM in our record fits those studies cited. Therefore, it is reasonable to shortly summarize the main interpretations of the previous work regarding the timing of the HTM. This is summer insolation and teleconnection with Europe. Deleting the teleconnection part would make the summary incomplete.

Lines 428-429: It is unclear what this sentence means. Please clarify.

The sentence will be deleted.

Fig. 2: Plot age and proxy uncertainty envelopes for all data from core 12KL. Proxy uncertainty includes analytical uncertainty (relatively small) and calibration uncertainty (quite large, relative to the signal). This is important to report.

We decided to implement the calibration error alone, since the analytical error is so small that it is hardly wider than the line of the plots.

Fig. 4: show the model LGM land boundaries and the PI land boundaries. Are these annual or summer anomalies? Clarify in the figure caption.

The figure shows boreal summer (JJA) anomalies. This will be implemented into the caption. Furthermore, we will include the land boundaries applied to the PI and LGM simulations into Figures 4a and 4b (see supplementary figures to this response letter):

Supplementary Figure 1 (this will replace Figure 4a in the manuscript). COSMOS-simulation for the JJA SLP-anomaly over Beringia and the N Pacific during the LGM (21 ka) relative to PI. Arrows represent the wind anomaly.

Supplementary Figure 2 (will replace figure 4b in the manuscript). COSMOS-simulation for the SAT-anomaly together with the wind-anomaly during JJA for LGM relative to PI conditions.

Technical corrections: Line 12: Branched Glycerol. . .does not need to be capitalized

Yes, thanks. Capital letters will be replaced by lower case letters.

Line 35: clarify what "next to" means: rather than? or in addition to?

It is meant in the sense of "in addition to". This will be replaced since "next to" seems to be a confusing term.

Line 53 (and elsewhere in the text): I think the authors mean 150°W here, and this same typo is made elsewhere (e.g., line 385).

Yes, there is a typo, thanks for pointing out. "150°E" is the correct term and will be included. The same applies to line 385 and a few more.

Line 74: clarify what "over the northern shelves of central Beringia" means. Is this a geographic location? Could this be highlighted on a map or described in more clear terms? This describes the geographic location of the average position of the EAT. If this is difficult to understand, we can write "over the Chukchi Shelf" instead of "shelves of central Beringia". As the Chukchi Shelf is indicated on the map in Figure 1, this should clarify.

Line 175: it might help to add the word "respectively" to the end of the sentence that lists the percentages. Will be done.

Line 184: change "with approx." to "at approx." Will be done.

Line 194: Add "North" before "American continent" Will be done.

Line 203: remove the w in "now" Yes, thanks.

Line 209: it might be more clear to say that the SAT anomaly becomes stronger or becomes more pronounces from east to west (because the anomaly is actually decreasing from east to west).

That is a good point, thanks. We will include "becomes more pronounced from east to west".

Line 226: this is an incomplete sentence.

Indeed. The sentence will be completed by: "...Ti/Ca indicates relatively high contributions of terrigenous material relative to marine components at 15.5 and 12 ka BP and relatively low terrigenous contributions at 14 and 11 ka BP."

Line 248: add comma between climate and according

Will be implemented.

Line 250: change 'computer' to 'general circulation'

Will be done.

Line 253: change "ice caps" to "ice sheets"

Will be done.

Line 255: is CO2atm defined prior to this? If not, then define here.

You are right, "CO2atm" appears for the first time. We will define it: "atmospheric CO2 (CO2atm)."

Line 261: add "summer" between present and conditions.

Will be done.

Line 412: define what "it" refers to.

"It" refers to the atmospheric linkage, which is mentioned in the previous line. In order to keep the sentence clear, we replaced it by "the linkage"

Line 420: change "were as high as at present" to "were similar to present temperatures" or something to that effect.

We suggest to add "were very similar to present summer temperatures"

Line 421: remove "a" before "stronger-than-present"

Will be done.

References (not listed in the paper): Braconnot, P., Harrison, S. P., Kageyama, M., Bartlein, P. J., Masson-delmotte, V., Abe-ouchi, A., . . . Zhao, Y. (2012). Evaluation of climate models using palaeoclimatic data. Nature Climate Change, 2(6), 417–424. http://doi.org/10.1038/nclimate1456 Dullo, W. C., B. Baranov, and C. van den Bogaard (2009), FS Sonne Fahrtbericht/Cruise Report SO201–2. IFM-GEOMAR, Rep. 35, 233 pp, IFM-GEOMAR, Kiel, Germany. Kuehn, H., L. Lembke-Jene, R. Gersonde, O. Esper, F. Lamy, A. Arz, G. Kuhn, and R. Tiedemann (2014), Laminated sediments in the Bering Sea reveal atmospheric teleconnections to Greenland climate on millennial to decadal timescales during the last deglaciation. Clim. Past, 10(6), 2215–2236, doi:10.5194/cp-10-2215-2014. Lund, D. C., A. C. Mix, and J. Southon (2011), Increased ventilation age of the deep northeast Pacific Ocean during the last deglaciation, Nat. Geosci., 4(11), 771–774, doi:10.1038/ngeo1272. Max, L., Riethdorf, J.-R., Tiedemann, R., Smirnova, M., Lembke-Jene, L., Fahl, K., Nürnberg, D., Matul, A. and Mollenhauer, G.: Sea surface temperature variability and sea-ice extent in the subarctic northwest Pacific during the past 15,000 years, Paleoceanography, 27(3), PA3213, doi:10.1029/2012PA002292, 2012. Meyer, V. D., Max, L., Hefter, J., Tiedemann, R. and Mollenhauer, G. Glacial-to-Holocene evolution of sea surface temperature and surface circulation in the subarctic northwest Pacific and the Western Bering Sea. Paleoceanography 31, (2016), doi:10.1002/2015PA002877. Praetorius, S. K., A. C. Mix, M. H. Walczak, M. D. Wolhowe, J. A. Addison and F. G. Prahl (2015), North Pacific deglacial hypoxic events linked to abrupt ocean warming. Nature, 527(7578), 362-366, doi:10.1038/nature15753. Raddatz, T. J., Reick, C. H., Knorr, W., Kattge, J., Roeckner, E., Schnur, R., Schnitzler, K.-G., Wetzel, P., and Jungclaus, J.: Will the tropical land biosphere dominate the climate–carbon cycle feedback during the twenty-first century?, Clim. Dynam., 29, 565–574, 2007. Reimer, P. J., et al. (2009), Intcal09 and Marine09 radiocarbon age calibration curves, 50,000 years cal. BP, Radiocarbon, 51(4), 1111–1150. Sarnthein et al., (2015). Planktic and Benthic 14C Reservoir Ages for Three Ocean Basins, Calibrated by a Suite of 14C Plateaus in the Glacial-to-Deglacial Suigetsu Atmospheric 14C record. Radiocarbon, Vol 57, 1, 2015, p 129–151

Staerz, M., G. Lohmann, and G. Knorr, 2016: The effect of a dynamic soil scheme on the climate of the mid-Holocene and the Last Glacial Maximum. Clim. Past 12, 151-170. doi:10.5194/cp-12-151-2016 Stuiver, M., and P. J. Reimer (1993), Extended C-14 Data-Base and Revised Calib 3.0 C-14 Age Calibration Program, Radiocarbon, 35(1), 215–230.

[Figure]

[Figure]

−2750  −2250  −1750  −1250  −750  −250  250  750  125

---

## Author Comment (AC2) · 4 Aug 2016

Dear reviewer, we appreciate your helpful suggestions to improve our manuscript very much. Please, see below how we would like to revise our manuscript.

Abstract Line 12: Rather than 'western continental margin off Kamchatka/marginal Northwest Pacific ' suggest 'western continental margin off Kamchatka in the Northwest Pacific'

Indeed, the sentence reads better without the slash. We will adopt your suggestion.

1.Introduction Line 32-37: Long, awkward sentence. Suggest reworking for clarity.

[Figure]

We suggest to rewrite the parapraph as follows: "Kamchatka is one of the least studied areas of Beringia. Since the 30 available terrestrial climate archives, such as peat sections or lake sediments, do not reach beyond 12 ka BP (e.g. Dirksen et al., 2013, 2015; Nazarova et al., 2013a; Hoff et al. 2015; Klimaschewski et al., 2015; Self et al., 2015; Solovieva et al., 2015). The climatic conditions during the LGM and the deglaciation are poorly understood."

Line 37: Unsure why this sentence begins with 'Particularly' in context of previous sentence.

"Particularly" is not needed and will be deleted.

On line 40: Statement that majority of sea surface temperature records from the subarctic NW Pacific and marginal seas mirror N. Atlantic climate oscillations needs to be qualified. I am assuming that this claim pertains to millennial scale N. Atlantic climate oscillations as recorded from ice core d18O? If so, Caissie et al. 2010 reference is for a surface ocean temperature record of multi-millennial scale resolution with very limited chronological control that bears little resemblance in structure to NGRIP d18O outside of a crude transition from apparently full glacial to interglacial conditions between 12-11 ka. Of the 6 records from the 'NW Pacific and marginal seas' presented by Max et al., 2010, while the nearly all the color b* records resemble deglacial NGRIP d18O in structure, only one of the attendant SST records (the NW Pacific core SO201-12-KL that is also the subject of this paper) looks anything like NGRIP d18O at millennial scales. I can't comment on the Meyer et al. reference as it's unpublished. If correct, this rather sweeping assertion that NW Pacific SST mirrors N. Atlantic climate has fairly important implications vis-a-vis the following suggestion that N Atlantic teleconnections control deglacial temperature development in the N. Pacific. However this assertion seems poorly defended by the references offered in the text at this point and at the very least needs further qualification/elaboration.

We agree that the statement appears not well defended as the reference list is incomplete. Therefore, we decided to include more references, also from the NE Pacific (Barron et al., 2003; Praetorius and Mix, 2014; Praetorius et al., 2015). Together with the references listed, the current state of the art will be well represented and the idea that the atmospheric teleconnections with the N-Atlantic were important for deglacial climate change in the N-Pacific realm, will be much better defended. The "NW-Pacific" in line 41 will be replaced by "North Pacific". The Meyer et al., submitted is now published (Meyer et al., 2016).

Line 44: In the marine environment you only address records from the broader NW Pacific, but then for terrestrial records you include records from the Alaskan portion of Beringia. Why is there no discussion of the well-dated marine records of climate from the NE Pacific, or alternatively, why are the terrestrial Alaskan records being included in this discussion?

See comment on line 40. We will include references for the SST development in the NE Pacific.

2. Regional Setting Line 71: Suggest replacing 'which are' with a comma.

Will be done.

Line 75: Why is Jet capitalized? Should it be 'westerly jet stream'? Similarly 'Jetstream' in line 78 should also be two lowercase words.

You are right, "Jet" does not need to be capitalized. This will be changed in both lines.

Line 82: Add comma after 'ranges'.

Will be done.

Line 84: Would read better as 'Mean temperatures averaged for the entire Peninsula range from: : :'. Alternatively, place a comma after 'Peninsula'.

That's true. We will write: "Mean temperatures averaged for the entire Peninsula. . ."

3. Materials and Methods 3.1 Core material and chronology Although you reference Max et al. (2012) for details of chronology, as it's highly pertinent to this paper it would be nice to have some basic information offered on the core length and the number of radiocarbon dates that constrain accumulation. Similarly, as these results are the subject of another paper, a mention of the mean sedimentation rates in the Holocene and deglacial sections of the core (properly cited) would be useful to the reader.

This information will be included into the paragraph: "Age control is based on accelerator mass spectrometry (AMS) radiocarbon dating of planktic foraminifera (Neogloboquadrina pachyderma sin; 9 dates in total) as well as on correlations of high-resolution spectrophotometric (color b*) and X-ray fluorescence (XRF) data of different sediment cores from the NW Pacific, the Bering Sea and the Sea of Okhotsk (Max et al., 2012). The correlation allowed to transfer AMS results from core to core, which provided 10 more age control points for site 12KL (Max et al., 2012). Based on the age model by Max et al. (2012) Holocene, deglacial and glacial sedimentation rates are 39, 79 and 59 cm/ka, respectively, allowing to investigate climate change on multicentennial to millennial timescales (Max et al., 2012). The core has a length of 11.78 m representing the past 20 ka (Dullo et al., 2009; Max et al., 2012). It was sampled in 10 cm steps providing an average resolution of approximately 200 years between samples (Meyer et al., 2016).

3.2 Lipid Extraction Line 101: No need for a comma after n-hexane.

Will be deleted

3.4 Temperature determination I'm unclear from this how the BIT-index controls for GDGT's from fresh water environments?

We will add the following sentence to line 144: "The higher BIT index values the larger the relative contributions from terrestrial soil or fresh-water sources."

In this and previous (3.3) section, in many locations in the text I'm unclear on what

precipitates the use of the abbreviation GDGT as opposed to brGDGT? I would have guessed it was branched versus all Glycerol Dialyl Glycerol Tetraethers, but in some cases (if I'm not mistaken) GDGT appears to be used interchangeably with brGDGT. Really this comment could extend to entire manuscript.

In section 3 "GDGT" is used for the total GDGT distribution comprising isoprenoid and branched GDGT. "brGDGT" is used when only brGDGT are intended to be addressed. If specific brGDGTs are mentioned (e.g. GDGT III) the "br" is not indicated in order to keep consistency with the common nomenclature in the literature. In order to clarify that the total GDGT pool is addressed, we may implement a sentence saying that isoprenoid and branched GDGT were isolated from the sediments and then abbreviate with GDGTtotal or GDGTiso+br.

In section 4. "GDGT" is indeed exchangeable with "brGDGT". Initially, we abbreviated "branched GDGT" by "GDGT". Later this was specified by "brGDGT" and all "GDGT"s were supposed to be replaced. It seems that some of them managed to hide from the change. We will correct this in all instances throughout the manuscript. Thanks for pointing out.

Line 145: pluralize 'sample'. Also no need for 'present'.

Will be done.

4. Results 4.1 Concentrations and fractional abundance of brGDGT It would seem to me that Figure 3 (and the discussion of it) should precede Figure 2 in the text.

We agree that there is reason to let Figure 3 precede Figure 2, because in sections 4 and 5 fractional abundances are discussed before the temperature evolution. However, in section 4 and 5 the concentrations of $\Sigma$brGDGT and the BIT-index are discussed prior to the fractional abundances (and the concentrations and BIT-index are shown in Figure 2). This order is required by the logic of section 4.1 where the low BIT index is the reason for a detailed discussion about the sources of brGDGT in core 12KL. The

fractional abundances are used to evaluate the sources. In order to keep a consistent organization throughout sections 4 and 5 we decided to introduce the results for concentrations (given in Figure 2) prior to the results for fractional abundances (Figure 3). This requires Figure 2 to precede Figure 3.

4.2 Temperature development over the past 20 ka

Line 178: This is a very narrow definition of the late Holocene (1 ka BP), and it only affords you one data point in the record of 12KL to compare to the dozen or so available for the 2 ka window afforded the glacial (18-20 ka BP). Perhaps consider broadening the definition of 'late Holocene' and when presenting a surface temperature include a standard deviation that encompasses both analytical/calibration error as well as observed variability over that time interval.

That is good point, thanks. We will broaden the window of the late Holocene to 1-3 ka BP. Temperature is on average $7.3 \pm 5.3°C$ during this interval, the same as between 18 and 20 ka BP.

Also, here and every future instance, why is approximately abbreviated?

Will be written out everywhere.

While I don't doubt that the glacial temperatures are statistically 'the same' as those in the late Holocene, this could easily be presented quantitatively.

See comment on line 178

Line 179: In this and every instance throughout the paper, when giving a temperature, present that value in the context of its uncertainty.

This will be done with all our MATifs values. As for literature data it will be added if the uncertainty estimates are given by the source of the data.

Line 179-182: This pair of sentences is awkward and read poorly. The discussion of the single warm data point at 16 ka reads like a stream of consciousness as opposed

to a well-digested scientific observation.

We will rewrite the section: "With the beginning of the deglaciation at 18 ka, temperature drops by about $1.5 \pm 5°C$. It remains colder than during the LGM until 14.6 ka BP where the temperature abruptly jumps back to the LGM-level."

Line 181: If you're going to present ages down to the century scale, you need to include estimates of temporal error that reflect the chronological control of the core.

We suggest to implement a discussion of age-model uncertainties into section 3.1 where the achievement of age control is described. "Max et al. (2012) converted radiocarbon ages into calibrated calendar ages using the calibration software Calib Rev 6.0 (Stuiver and Reimer, 1993) with the Intcal09 atmospheric calibration curve (Reimer et al., 2009). A constant reservoir age of 900 years was assumed for the entire time-interval covered by the core (Max et al., 2012). The uncertainty of AMS dating was smaller than $\pm$ 100 years (Max et al., 2012). Another important issue are changes in reservoir ages of the surface ocean, in particular during the last deglaciation (Sarnthein et. al., 2015). However, recent studies suggest that reservoir ages of the Bering Sea and the N-Pacific varied by less than 200 years during the last deglaciation (Lund et al., 2011; Kühn et al., 2014) and are within the range of reservoir ages originally assumed by Max et al. (2012)."

Line 187: What is the average mid-Holocene thermal maximum temperature between 8.0-4.0 ka BP (with errors)? Perhaps present the average temperature in that window, then give the highest temperature reached and the age (again, with errors) that that peak temperature is observed.

The sentences in lines 186-189 will be replaced by: "After the abrupt temperature increase into the Preboreal temperature increases progressively rises culminating into a Mid-Holocene Thermal Maximum between 8.0 and approximately 4.0 ka BP (average temperature $7.7 \pm 5.2°C$)."

Line 188: How do you determine when the cooling trend is initiated? It would seem that the cooling arguably begins closer to 5 ka, but then again if you're interpreting at this level (and you probably shouldn't) you could argue the cooling stops by 3 ka.

As the term Holocene thermal maximum already implies that temperature is lower in the periods preceding and following the HTM, the last sentence will be deleted.

Line 189: This last sentence needs to be quantitative. Also, when calculating variance, remember to use equivalent temporal windows for the Holocene and deglacial and smooth the record to a constant resolution.

Actually, the last sentence is trivial since it is well known that the Holocene climate variability is generally rather constant compared to the one of the deglaciation. Furthermore, the main findings of this paper are that LGM summers were similarly warm as during the Holocene and that the Greenland like millennial-scale oscillations characterize the deglaciation. So, the sentence in line 189 is not important for the paper and we will delete it.

5. Discussion 5.1 Sources of brGDGT and implications for CBT/MBT'-derived temperatures Line 219: Either here or in the methods section some very basic discussion of how to interpret the BIT-values should be given.

See comment on "3.4. Temperature determination"

Line 221: Eminent might not be the right word choice for this sentence. Perhaps 'Marine settings where terrigenous input is low are particularly sensitive to bias from in-situ production, thus non-soil derived brGDGTs potentially have a considerable effect on the temperature: : :'

We will implement the sentence you suggest.

Line 225: Again, would suggest minor reworking. Perhaps 'Ti/Ca-ratios reflect the proportion of terrigenous and marine derived inorganic components of the sediment, and can be used as an estimator of terrigenous input'.

That reads better, thanks.

Line 226: 'With relatively high values at 15.5 and 12 ka BP, and minima at 14 and 11 ka BP' is an incomplete sentence. Also, again, if presenting chronologies at the centennial scale really need to give errors on those ages.

Indeed, the sentence will be completed by: . . .Ti/Ca indicates relatively high contributions of terrigenous material relative to marine components at 15.5 and 12 ka BP and relatively low terrigenous contributions at 14 and 11 ka BP.

For chronology see comment on line 181.

Line 244: 'Mai' should be 'May'.

Thanks, will be changed.

5.2 Temperature evolution over the past 20 ka 5.2.1 The LGM (20-18 ka) – warm summers and the regional context What definition of LGM are you using? Should give a reference. Clark et al., 2009 is the most widely used citation that I'm aware of and they define global LGM as ending at 19 ka.

We are referring the definition of Mix et al. (2001) according to which the LGM lasted from 18-24 ka BP.

Line 251: While you could say there was a 'cooling tendency' from MIS-3 into the LGM, since time moves forward when comparing the LGM to the Holocene it would be better to say 'Generally cooler LGM temperatures are thought to result from: : :'

That is a good point, thank you. We will rearrange the sentence as you suggest.

Line 257: What does 'BLB' stand for?

BLB stands for "Bering Land Bridge". This is defined in the introduction (line 29). We will repeat the definition to make the text easier to understand.

Line 260: No need to hyphenate 'insect-data'. Also suggest rewording to 'Markovo,

and ElGygytgyn and Jack London lakes'

Will be changed.

5.2.2. Controls on MATifs In this section you identify a possible seasonal bias in alkenone-based SST reconstructions towards warmer temperatures and dismiss them in favor of TEXL86 reconstructions. You then discuss the results of the TEXL86 reconstruction for site 12KL currently submitted for review. However there is no discussion of the already-published alkenone-based SST record for 12KL of Max et al., (2012), nor is there a presentation of this record alongside the TEXL86 record from the same site in Figure 2. For the period of overlap, it would appear that at least at this location the alkenone SST's are several degrees colder than the TEXL86 temperature reconstruction. Why would this be?

The alkenone temperature record from site 12KL is excluded from the LGM discussion since it does not reach beyond 16 ka BP (see Max et al., 2012). For the discussion on millennial scale oscillations we did not show the UK'37 as it is in line with the TEXL86 from core 12KL (Meyer et al., 2016). Since the general trend is the same in both records it appears more reasonable to represent the SST evolution of the NW Pacific by the TEX which spans the entire LGM-Holocene transition. Differences between UK'37 and TEXL86 are discussed in Meyer et al., 2016 (in the paper referenced as "Meyer et al., submitted") and are attributed to different blooming seasons of coccolithophores and archaea.

Line 266: Need to clarify that you're discussing warm Siberian summers during LGM

You are right. Thanks. This will be changed.

Line 288: As the paper Meyer et al., (submitted) has yet to pass through peer review, probably best to state that the relatively warm SST's at site 12KL may be explained by stronger-than-present influence of the Alaskan Stream.

Since the Meyer et al (submitted) is now accepted (Meyer et al., 2016), nothing will be

changed.

5.2.3 The deglaciation (18 ka-10 ka BP) Define/defend the use of the words 'strong' and 'clear' when describing the resemblance between the N-Atlantic d18O and 12KL MAT. Can you calculate covariance between the normalized/equivalently smoothed NGRIP d18O and 12KL MAT? To my eye they appear quite different: the Y-D is greatly compressed in the Kamchatka MAT record, the trend from the LGM to HS1 in 12KL is completely absent in NGRIP. A climate oscillation in HS1 apparently comparable in magnitude and duration to the regional expression of the Y-D (although a warming as opposed to cooling event) at 16 ka with no analogue in NGRIP is discarded from interpretation. I'm not arguing that there are similarities, but to say it's obvious or 'undoubtable' that the North Atlantic is driving NW Pacific climate via atmospheric teleconnection is a strong claim that needs to be quantitatively defensible. If this can't be done in the context of this paper, perhaps dial the tone of the text down a bit. Also, as stated earlier in the text, when comparing 12KL to NGRIP at centennial scales chronological uncertainties in 12KL need to be addressed and stated.

We will "delete" clear and "strong" and "undoubtedly" As for age uncertainties, see comment on line 181.

5.2.4 The Holocene The statement at line 411: "Hence it seems that the atmospheric linkage (with the N Atlantic) that determined climate variability during the deglaciation likely persisted into the Holocene where it acted as an important driver for long-term climate changes as well as abrupt, short-lived climate events." seems poorly defended by the visual similarity between NGRIP d18O and Kamchatka MAT in Figure 2. To my eye the Holocene in the MAT record appears more variable, while the mid-Holocene thermal maximum and neoglacial cooling described for the NW Pacific region are absent in NGRIP. Quantitatively evaluating the covariance between these records would be challenging at best as the current chronology for 12KL is virtually unconstrained in the Holocene. If this statement remains in the discussion/conclusions, at the very least some discussion of what is meant by 'long-term climate changes' versus 'abrupt,

short-lived climatic events'.

We will delete the sentences in lines 409-413 as the presence of an 8.2 event on Kamchatka is debatable on the basis of the existing records and it is not apparent in MATifs.

6. Summary and Conclusions Line 415-419: This introduction to the conclusions reads awkwardly.

We will delete lines 417-419 and also the listing (i; ii) in the following.

Line 420: Perhaps replace 'likely' with 'may' or 'could' as there is no evaluation of statistical certainty of this hypothesis.

Will be replaced by "may".

Line 433: Again, the use of the word 'obvious' to describe the role of N-Atlantic climate in driving the NW Pacific seems somewhere between bombastic and unfounded. There are some similarities in deglacial climate, there are differences, and as yet these remain poorly quantified in the manuscript.

"Obvious" will be replaced by "seem to be linked".

Figures

Figure 1: Could some kind of shading be used to more clearly denote Holocene land-masses? With apparently identical solid lines used to denote boundaries of continents, ocean currents, and rivers it's a bit difficult to visually parse.

In order to increase the contrast between land and ocean we will add a grey shading to the Holocene land masses.

Figure 2: As this figure includes the TEXL86 SST record from Site 12KL to be published in Meyer et al., submitted, it should probably also include the deglacial alkenone SST record from site 12KL published in Max et al., 2012.

See comment on section 5.2.2 Controls on MATifs

Figure 3: As mentioned in my comments on the results section, I think this figure should be reversed with Figure 2 in its presentation order in the text. Also, instead of giving ages at 4 depths in the core, could a secondary axis with appropriately dilated/ compressed ticks be added for age alongside the depth scale? If this isn't possible, would almost suggest it would be better to present results versus time than versus depth to facilitate comparison to Figure 2.

As discussed above (see comment on the result section), we would rather leave the order as is.

We will increase the density of the age scale in Figure 3.

References:

Barron, J. A., L. Heusser, T. Herbert, and M. Lyle (2003), High-resolution climatic evolution of coastal northern California during the past 16,000 years. Paleoceanography, 18(1), 1020, doi:10.1029/2002PA000768. Dullo, W. C., B. Baranov, and C. van den Bogaard (2009), FS Sonne Fahrtbericht/Cruise Report SO201–2. IFM-GEOMAR, Rep. 35, 233 pp, IFM-GEOMAR, Kiel, Germany. Kuehn, H., L. Lembke-Jene, R. Gersonde, O. Esper, F. Lamy, A. Arz, G. Kuhn, and R. Tiedemann (2014), Laminated sediments in the Bering Sea reveal atmospheric teleconnections to Greenland climate on millennial to decadal timescales during the last deglaciation. Clim. Past, 10(6), 2215–2236, doi:10.5194/cp-10-2215-2014. Lund, D. C., A. C. Mix, and J. Southon (2011), Increased ventilation age of the deep northeast Pacific Ocean during the last deglaciation, Nat. Geosci., 4(11), 771–774, doi:10.1038/ngeo1272. Max, L., Riethdorf, J.-R., Tiedemann, R., Smirnova, M., Lembke-Jene, L., Fahl, K., Nürnberg, D., Matul, A. and Mollenhauer, G.: Sea surface temperature variability and sea-ice extent in the subarctic northwest Pacific during the past 15,000 years, Paleoceanography, 27(3), PA3213, doi:10.1029/2012PA002292, 2012. V. D. Meyer, V,. D., Max, L., Hefter, J., Tiedemann, R. and Mollenhauer, G. Glacial-to-Holocene evolution of sea surface temperature and

surface circulation in the subarctic northwest Pacific and the Western Bering Sea. Pale-oceanography 31, (2016), doi:10.1002/2015PA002877. Mix, A. C., Bard, E., Schneider, R. (2001). Environmental processes of the ice age: land, oceans, glaciers (EPILOG). Quat. Sci. Rev. 20, 627-657. Praetorius, S. K., and A. C. Mix (2014), Synchronization of North Pacific and Greenland climates preceded abrupt deglacial warming. Science, 345(6196), 444-448, doi:10.1126/science.1252000. Praetorius, S. K., A. C. Mix, M. H. Walczak, M. D. Wolhowe, J. A. Addison and F. G. Prahl (2015), North Pacific deglacial hypoxic events linked to abrupt ocean warming. Nature, 527(7578), 362-366, doi:10.1038/nature15753. Reimer, P. J., et al. (2009), Intcal09 and Marine09 radio-carbon age calibration curves, 50,000 years cal BP, Radiocarbon, 51(4), 1111–1150. Sarnthein et al., (2015). Planktic and Benthic 14C Reservoir Ages for Three Ocean Basins, Calibrated by a Suite of 14C Plateaus in the Glacial-to-Deglacial Suigetsu Atmospheric 14C record. Radiocarbon, Vol 57, 1, 2015, p 129–151 Stuiver, M., and P. J. Reimer (1993), Extended C-14 Data-Base and Revised Calib 3.0 C-14 Age Calibration Program, Radiocarbon, 35(1), 215–230.

---

## Author Response (AR1)

Dear Dr. Kiefer,

thank you very much for asking us to submit a revised version of our manuscript. Please, find below how we finally addressed your comments as well as the reviewer's comments in this new version of the article. Your comments and the reviewer's comments are written in black, our replies are blue. All line numbers refer to the tracked-changes version of the article.

In addition to suggested changes by the reviewers we rearranged the introduction (section 1) and parts of the discussion in order to improve the logic and the flow of the text. Also, most parts of section 3.2 (which describes the lipid extraction procedure) were replaced by a citation (Meyer et al., 2016).

Yours sincerely, Vera Meyer

**Reply to Editor's comments:**

I do notice that you responded to Ref#2's question about the selection of records taken into consideration and their geographical distribution also simply by adding references. I wonder whether this will self-explain your reasoning or whether some formulated explanations wouldn't be more appropriate.

We agree with the reviewer's concern that if the terrestrial records from Alaska are included the marine records from the NE Pacific should be mentioned as well. Therefore, we added references for records from the NE Pacific. In order to clarify that both the NW Pacific and NE Pacific are meant we write "N Pacific".

As a minor editorial point, I would ask you to consider minimising abbreviations in the text, especially local ones, such as BLB and AS.

Full expressions of BLB and AS are given.

**Reply to reviewer #1**

Dear reviewer,

thank you very much for your detailed and constructive review on our manuscript. Please, find below how we addressed your comments in the revised version. Your comments are given in black, our replies in blue.

**Specific comments**

Line 166: list the number of samples analyzed and the approximate or average depth and time sampling resolution Done.

Line 204: How exactly was the standard deviation measured? Is this for this lab or labs in general? Was a standard measured regularly among sample injections, or were the repeat measurements of the samples? What is the pooled standard deviation of samples that were run

multiple times (if any?...and if none, then that is important to report)? Please clarify in the text.

The standard deviation derives from repeated measurements of a standard sediment extract (n=7) which had been treated in the same way as the samples. Repeated measurement for the samples of core 12KL were not possible, because of small amounts. We mention the standard sediment in line 205.

Line 215: it is critical to report the calibration error, which is much larger than the analytical error. While relative changes in a single record are likely real, the absolute temperature change is difficult to pinpoint because of this large calibration error. Please clarify in the text. The calibration error ( $\pm$  5°C) as well as the analytical uncertainty of our measurements is noted in section 3.4. We agree with the reviewer that absolute changes are difficult to pinpoint but that relative changes in the records are real and that the latter make the important information of the data presented in our study. Therefore, whenever absolute estimates of temperature changes appear in the text they are presented with a ~. (e.g. ~7.5°C).

Line 230: does the word "glacial" belong here? it doesn't make sense. Please clarify in the text.

No, the word does not belong here. Deleted. Thanks!

Line 238-239: what parameters were used in this simulation? While the reader can check the references given here, it would be good to briefly summarize the parameters that were used to force the model and the parameters that were changed between the glacial and preindustrial runs (ice volume? sea level? orbital forcing? Greenhouse gases? land cover? etc: : :.) Please clarify in the text. Ice volume, sea level, and land cover are of particular importance for this region, where a large amount of land was exposed during the LGM.

The following paragraph is included:

"External forcing and boundary conditions are imposed according to the protocol of PMIP3 for the LGM (available at http://pmip3.lsce.ipsl.fr/). The respective boundary conditions for the LGM comprise orbital forcing, greenhouse gas concentrations ( $CO_2=185$ ppm;  $N_2O=200$ ppb;  $CH_4=350$ ppb), ocean bathymetry, land surface topography, run-off routes according to PMIP3 ice sheet reconstruction and increased global salinity (+ 1 psu compared to modern value) to account for a sea-level drop of ~116 m. The glacial ocean was generated through an ocean-only phase of 3000 years and coupled phase of 3000 years (LGMW in Zhang et al., 2013). The land cover is calculated interactively in the climate model which has an interactive land surface scheme and vegetation module (Brovkin et al. 2009). The modular land surface scheme JSBACH (Raddatz et al., 2007) with vegetation dynamics (Brovkin et al., 2009) is embedded in the ECHAM5 atmosphere model. The background soil characteristics (which are described in Staerz et al., 2016) are set to the values which are closest to the preindustrial land points."

Line 253: what does "integrated" mean? is this the model spin up? was this done twice, once for each the LGM and Pre-Industrial runs? Please clarify in the text. Integration means "simulated years".

In order to clarify we write: "For both, PI and LGM conditions, the climate model was integrated twice for 3000 model years and provides monthly output (Wei et al., 2012; Wei and Lohmann, 2012; Zhang et al., 2013)."

Line 282: Because this record is not in the North Atlantic, it would be best to avoid using terms that are related to North Atlantic climate change (ie. Bolling Allerod) in this resultsoriented portion of the paper. When the authors later discuss links with the North Atlantic, these North-Atlantic-based terms can and should be introduced.

We replace Atlantic-related terms by dates. For example: "...until the beginning of the Bølling/Allerød" turns into:"...until 14.6 ka BP."

Line 311: It seems to me that the change from exposed land during the LGM to ocean during the PreIndustrial run over the land bridge would be a source of large changes in modeled SAT. This aspect is important to address, not only how this is handled in the model (is this exposed land in the LGM simulation?), but also how this could affect SAT in the model, and whether that is similar to the real-world effects. I would question whether these anomalies are even meaningful, and would need more explanation of what the changes mean, because of the changes from land to ocean surface.

As already mentioned in the reply to the comment on line 238-239, we add a paragraph describing the model setup in more detail. The effects of the exposed land in the model and the proxy world are addressed in the discussion (lines 505-520). Moreover, we show land boundaries for LGM and PI conditions. This clarifies that these anomalies occur over exposed land in the model.

Line 341: using slashes to indicate opposite effects is confusing. I suggest removing them and adding a phrase at the end of the sentence, like "with the opposite effect occurring with low terrigenous input". See [Robock, 2010] for a humorous take on how confusing it can be to use slashes to express opposites.

Lines 330-340 are rewritten and the slashes have been removed.

Line 346: do the authors mean "in marine areas where brGDGTs are thought..."? Please clarify.

Yes. Sentence is rephrased.

Line 360: what are the uncertainties or standard deviations on these temperature observations? Please clarify.

The calibration error is  $\pm 5^{\circ}$ C which is noted in section 3.4. In order to remind the reader at this point in the text we add the error to the CBT/MBT' temperature estimates in those lines.

Line 368: cite PMIP? Done.

Line 390: clarify whether this attribution was by previous studies, or by this study. By previous studies. Text clarified.

Line 404 and others: Clarify in the text what proxy was used to produce this Sea of Okhotsk SST reconstruction.

The sentence encompassing lines 404 and 405 begins with: Alkenone-based SST... This implies the application of  $U^{K'}_{37}$ . In order to specify we replaced "Alkenone-based" by " $U^{K'}_{37}$ ". In all other instances within lines 404-412 we mention the TEXL86 proxy, now.

Line 411: 1°C is well within calibration error of these proxies, and is important to mention in the text.

As the reviewer pointed out in his comment on line 215 relative changes in records based on those proxies are likely real and significant but absolute values of those changes are difficult

to pinpoint due to large calibration errors and due to different types of calibrations. However, the important point of the paragraph encompassing lines 400-410 is that at site 12KL the relative SST change is smaller than in the marginal seas when the same proxy and the same regional calibration are applied. Therefore, this regional difference is significant although the 1°C change at site 12KL is well within the calibration error of TEXL86 (1.7°C Seki et al., 2014). We do not change the paragraph.

Line 454-458: It seems as if the final two sentences in this paragraph say opposite things. Can this be clarified?

In order to clarify we append an additional paragraph to line 458.

Line 480: How robust or meaningful is this warmer-than-present temperature, given that there were large changes in surface conditions (land to ocean) from the LGM to present? I would expect summer temperatures to be quite warm over land, as dark soils can retain quite a bit of heat, whereas sea water remains much cooler. It is important to address the changing surface conditions in the text.

We include the land boundaries used in the PI and LGM simulations into Figure 4a and 4b (as requested in the reviewer's comment on Figure 4). Based on these modifications we also reorganized and expanded sections 5.2.2 and 5.2.2.1.

Line 531: It might clarify to add the following wording: "potentially explain the mismatch between model and proxy..." Changed.

Line 540: does the term "in the surrounding seas" refer to the Pacific or the Atlantic? It is unclear, as both are mentioned in this section. Please clarify.

It refers to the surrounding seas of Kamchatka, i.e. the Bering Sea, the subarctic NW Pacific and the Sea of Okhotsk. We put "in the surrounding seas of Kamchatka (Bering Sea, NW Pacific, Sea of Okhotsk)" along this line.

Line 553: in addition to the age model error, the authors must also discuss error in marine reservoir corrections, and it would be helpful show age uncertainties in Fig. 2 time series. This is an important point, thanks. In section 3.1 we add a short paragraph describing the uncertainties introduced by the AMS dating as well as by the assumptions for reservoir ages.

In line 553, we refer back to this paragraph.

Concerning age uncertainty estimates in Figure 2, we do not include any graphical features, e.g. bars, as those would make the figure look very busy. The information on uncertainty is given in section 3.1.

Line 562: clarify what proxy is used to reconstruct SST in the NW Pacific. Paragraph rewritten to clarify the content. No proxies are mentioned here.

Line 562-566: this sentence is unclear, please rewrite and clarify. Lines 558-570 are rewritten.

Line 568: What does AS stand for? Perhaps just spell out the full term. Thanks, "AS" stands for "Alaskan Stream". We decided to use the full term instead of the abbreviation everywhere in the text (see Editor's comments).

Line 570: why would ocean waters place a "restriction" on atmospheric teleconnections?

Can this be clarified?

This has been elaborated in Meyer et al., 2016 on the basis of two SST records from the Western Bering Sea and the NW Pacific. Lines 558-570 are rewritten

Line 613: I don't understand how the HTM is delayed on Kamchatka relative to other parts of Siberia, as both have the same beginning time (9ka). Can this point either be deleted or made more clear?

Since our paper focusses on the deglaciation and the LGM, the Holocene section was shortened. The lines encompassing the delay of the HTM are removed. For the same reason the pollen record from Kamchatka is removed from Figure 2.

Lines 620-623: these speculative connections with the North Atlantic seem like a stretch and could be explained by other, regional climate forcing mechanisms. Perhaps it would be best to remove these sentences?

These lines are removed.

Lines 638: It is unclear what this sentence means. Please clarify. The sentence deleted.

Fig. 2: Plot age and proxy uncertainty envelopes for all data from core 12KL. Proxy uncertainty includes analytical uncertainty (relatively small) and calibration uncertainty (quite large, relative to the signal). This is important to report.

Since the calibration error is relatively large it would require a large y-axis for the MATifs record. As a result the figure would probably become too large to fit on one page. Since the calibration error as well as analytical uncertainty are reported in section 3.4. nothing is changed.

Fig. 4: show the model LGM land boundaries and the PI land boundaries. Are these annual or summer anomalies? Clarify in the figure caption.

The figure shows boreal summer (JJA) anomalies. This is added to the caption. Furthermore, we include the land boundaries applied to the PI and LGM simulations into Figures 4a and 4b:

Figure 4a) COSMOS-simulation for the JJA SLP-anomaly over Beringia and the N Pacific during the LGM (21 ka) relative to PI. Arrows represent the wind anomaly.

4b) COSMOS-simulation for the SAT-anomaly together with the and wind-anomalies during JJA

**Technical corrections:**

Line 12: Branched Glycerol...does not need to be capitalized Done.

Line 44: clarify what "next to" means: rather than? or in addition to? It is meant in the sense of "in addition to". "Next to" is replaced by "in addition to" (see line 108)

Line 59 (and elsewhere in the text): I think the authors mean 150°W here, and this same typo is made elsewhere (e.g., line 584).

Yes, there is a typo, thanks for pointing out. "150°E" is the correct term. Changed everywhere.

Line 128: clarify what "over the northern shelves of central Beringia" means. Is this a geographic location? Could this be highlighted on a map or described in more clear terms? This describes the geographic location of the average position of the EAT. In the revised version we write "over the Chukchi Shelf" instead of "shelves of central Beringia". As the Chukchi Shelf is indicated on the map in Figure 1, this should clarify.

Line 270: it might help to add the word "respectively" to the end of the sentence that lists the percentages.

Done.

Line 281: change "with approx." to "at approx." Done.

Line 293: Add "North" before "American continent" Done.

Line 303: remove the w in "now" Done.

Line 311: it might be more clear to say that the SAT anomaly becomes stronger or becomes more pronounces from east to west (because the anomaly is actually decreasing from east to west).

Done.

Line 333: this is an incomplete sentence. Indeed. Sentence completed.

Line 363: add comma between climate and according Done.

Line 365: change 'computer' to 'general circulation' Done.

Line 369: change "ice caps" to "ice sheets" Done.

Line 371: is CO2atm defined prior to this? If not, then define here. Done

Line 379: add "summer" between present and conditions. Done.

Line 412: define what "it" refers to. Sentence removed.

Line 622: change "were as high as at present" to "were similar to present temperatures" or something to that effect. Done.

Line 623: remove "a" before "stronger-than-present"

Done.

**Reply to reviewer #2**

Dear reviewer,

Thank you very much for constructive and detailed review. Please find below how your comments are addressed in the revised version of our manuscript. Your comments are given in black, our replies in blue.

Abstract Line 14: Rather than 'western continental margin off Kamchatka/marginal Northwest Pacific ' suggest 'western continental margin off Kamchatka in the Northwest Pacific'

Changed

1. Introduction Line 30-35: Long, awkward sentence. Suggest reworking for clarity. Sentence has been rewritten (see lines 103-110).

Line 40: Unsure why this sentence begins with 'Particularly' in context of previous sentence. During the reorganization of the introduction, the sentence has been removed.

On line 42: Statement that majority of sea surface temperature records from the subarctic NW Pacific and marginal seas mirror N. Atlantic climate oscillations needs to be qualified. I am assuming that this claim pertains to millennial scale N. Atlantic climate oscillations as recorded from ice core d180? If so, Caissie et al. 2010 reference is for a surface ocean temperature record of multi-millennial scale resolution with very limited chronological control that bears little resemblance in structure to NGRIP d180 outside of a crude transition from apparently full glacial to interglacial conditions between 12-11 ka. Of the 6 records from the 'NW Pacific and marginal seas' presented by Max et al., 2010, while the nearly all the color b\* records resemble deglacial NGRIP d180 in structure, only one of the attendant SST records (the NW Pacific core SO201-12-KL that is also the subject of this paper) looks anything like NGRIP d180 at millennial scales. I can't comment on the Meyer et al. reference as it's unpublished.

If correct, this rather sweeping assertion that NW Pacific SST mirrors N. Atlantic climate has fairly important implications vis-a-vis the following suggestion that N Atlantic teleconnections control deglacial temperature development in the N. Pacific. However this assertion seems poorly defended by the references offered in the text at this point and at the very least needs further qualification/elaboration.

We agree that the statement appears not well defended as the reference list is incomplete. Therefore, we added more references, also from the NE Pacific (Barron et al., 2003; Praetorius and Mix, 2014; Praetorius et al., 2015), into the introduction and also into the discussion. Together with the references listed, the current state of the art will be well represented. In doing so, the idea that the atmospheric teleconnections with the N-Atlantic were important for deglacial climate change in the N-Pacific realm, will be much better defended.

Note that during the reorganization of the introduction this paragraph was shortened (see lines 89-100). This was done as it contained too much details which are all repeated in the discussion.

The Meyer et al., submitted is now published (Meyer et al., 2016).

Line 45: In the marine environment you only address records from the broader NW Pacific, but then for terrestrial records you include records from the Alaskan portion of Beringia. Why is there no discussion of the well-dated marine records of climate from the NE Pacific, or alternatively, why are the terrestrial Alaskan records being included in this discussion?

See comment on line 42. We will include references for the SST development in the NE Pacific and refer to the N Pacific as a whole. (see lines 87-90).

2. Regional Setting Line 126: Suggest replacing 'which are' with a comma. Done.

Line 131: Why is Jet capitalized? Should it be 'westerly jet stream'? Similarly 'Jetstream' in line 134 should also be two lowercase words. Changed.

Line 140: Add comma after 'ranges'. Done.

Line 142: Would read better as 'Mean temperatures averaged for the entire Peninsula range from: : :'. Alternatively, place a comma after 'Peninsula'. Changed.

3. Materials and Methods 3.1 Core material and chronology

Although you reference Max et al. (2012) for details of chronology, as it's highly pertinent to this paper it would be nice to have some basic information offered on the core length and the number of radiocarbon dates that constrain accumulation. Similarly, as these results are the subject of another paper, a mention of the mean sedimentation rates in the Holocene and deglacial sections of the core (properly cited) would be useful to the reader.

This information will be included into the paragraph: "Age control is based on accelerator mass spectrometry (AMS) radiocarbon dating of planktic

Age control is based on accelerator mass spectrometry (AWS) radiocarbon dating of plankte foraminifera (Neogloboquadrina pachyderma sin; 9 dates in total) as well as on correlations of high-resolution spectrophotometric (color b\*) and X-ray fluorescence (XRF) data of different sediment cores from the NW Pacific, the Bering Sea and the Sea of Okhotsk (Max et al., 2012). The correlation allowed to transfer AMS results from core to core, which provided 10 more age control points for site 12KL (Max et al., 2012).

Based on the age model by Max et al. (2012) Holocene, deglacial and glacial sedimentation rates are 39, 79 and 59 cm/ka, respectively, allowing to investigate climate change on multicentennial to millennial timescales (Max et al., 2012). The core has a length of 11.78 m representing the past 20 ka (Dullo et al., 2009; Max et al., 2012).

It was sampled in 10 cm steps providing an average resolution of approximately 200 years between samples (Meyer et al., 2016).

**3.2 Lipid Extraction Line 176: No need for a comma after n-hexane.**

Section has been shortened by citing Meyer et al (2016) for sample processing as we used the same extracts as these authors.

**3.4 Temperature determination**

I'm unclear from this how the BIT-index controls for GDGT's from fresh water environments?

The BIT-index is a means to estimate the relative abundance of brGDGT (terrigenous) vs isoprenoid GDGTs (marine) in marine sediments and is used to estimate terrigenous input to the sediments. It does not distinguish between soil or fresh water derived GDGTs. Nothing is done here, since the BIT index is interpreted in section 5.1.

In this and previous (3.3) section, in many locations in the text I'm unclear on what precipitates the use of the abbreviation GDGT as opposed to brGDGT? I would have guessed it was branched versus all Glycerol Dialyl Glycerol Tetraethers, but in some cases (if I'm not mistaken) GDGT appears to be used interchangeably with brGDGT.

Really this comment could extend to entire manuscript.

In section 3 "GDGT" is used for the total GDGT distribution comprising isoprenoid and branched GDGT. "brGDGT" is used when only brGDGT are intended to be addressed. If specific brGDGTs are mentioned (e.g. GDGT III) the "br" is not indicated in order to keep consistency with the common nomenclature in the literature.

In section 4. "GDGT" is exchangeable with "brGDGT", indeed. In the revised version "brGDGT" is used everywhere.

Line 225: pluralize 'sample'. Also no need for 'present'. Done.

4. Results 4.1 Concentrations and fractional abundance of brGDGT

It would seem to me that Figure 3 (and the discussion of it) should precede Figure 2 in the text.

We agree that there is reason to let Figure 3 precede Figure 2, because in sections 4 and 5 fractional abundances are discussed before the temperature evolution. However, in section 4 and 5 the concentrations of  $\Sigma$ brGDGT and the BIT-index are discussed prior to the fractional abundances (and the concentrations and BIT-index are shown in Figure 2). This order is required by the logic of section 4.1 where the low BIT index is the reason for a detailed discussion about the sources of brGDGT in core 12KL. The fractional abundances are used to evaluate the sources. In order to keep a consistent organization throughout sections 4 and 5 we decided to introduce the results for concentrations (given in Figure 2) prior to the results for fractional abundances (Figure 3). This requires Figure 2 to precede Figure 3.

4.2 Temperature development over the past 20 ka

Line 277: This is a very narrow definition of the late Holocene (1 ka BP), and it only affords you one data point in the record of 12KL to compare to the dozen or so available for the 2 ka window afforded the glacial (18-20 ka BP). Perhaps consider broadening the definition of 'late Holocene' and when presenting a surface temperature include a standard deviation that encompasses both analytical/calibration error as well as observed variability over that time interval.

Sentence was rephrased without giving age ranges for either the LGM or the Holocene. We do not add any uncertainties to absolute values in the text but mention the calibration error as well as the analytical uncertainties in section 3 (methods). All absolute values are given as approximations (e.g.  $\sim$ 7.5°C) and this does not necessarily require to present the value together with the uncertainty. Also, the analytical error (0.1°C), or errors imparted by variability over a time interval (0.3°C), are negligible compared to the calibration error (5°C). So, the order of magnitude of the uncertainties would be the same for every value ( $\sim$ 5°C).

Therefore, it should be sufficient to mention the calibration and analytical errors once in the method section.

Also, here and every future instance, why is approximately abbreviated? Done in every instance. Sometimes replaced by ~.

While I don't doubt that the glacial temperatures are statistically 'the same' as those in the late Holocene, this could easily be presented quantitatively. See comment on line 277.

Line 279: In this and every instance throughout the paper, when giving a temperature, present that value in the context of its uncertainty. See comment on line 277.

Line 279-282: This pair of sentences is awkward and read poorly. The discussion of the single warm data point at 16 ka reads like a stream of consciousness as opposed to a well-digested scientific observation. Section is rewritten.

Line 283: If you're going to present ages down to the century scale, you need to include estimates of temporal error that reflect the chronological control of the core.

We include a discussion of age-model uncertainties into section 3.1 where the achievement of age control is described.

"Max et al. (2012) converted radiocarbon ages into calibrated calendar ages using the calibration software Calib Rev 6.0 (Stuiver and Reimer, 1993) with the Intcal09 atmospheric calibration curve (Reimer et al., 2009). A constant reservoir age of 900 years was assumed for the entire time-interval covered by the core (Max et al., 2012). The uncertainty of AMS dating was smaller than  $\pm$  100 years (Max et al., 2012). Another source of uncertainty are changes in reservoir ages of the surface ocean, in particular during the last deglaciation (Sarnthein et. al., 2015). However, recent studies suggest that reservoir ages of the Bering Sea and the N-Pacific varied by less than 200 years during the last deglaciation (Lund et al., 2011; Kühn et al., 2014) and are within the range of reservoir ages originally assumed by Max et al. (2012)."

Line 289: What is the average mid-Holocene thermal maximum temperature between 8.0-4.0 ka BP (with errors)? Perhaps present the average temperature in that window, then give the highest temperature reached and the age (again, with errors) that that peak temperature is observed.

Added.

Line 291: How do you determine when the cooling trend is initiated? It would seem that the cooling arguably begins closer to 5 ka, but then again if you're interpreting at this level (and you probably shouldn't) you could argue the cooling stops by 3 ka. As the term Holocene thermal maximum already implies that temperature is lower in the

periods preceding and following the HTM, the last sentence is deleted.

Line 292: This last sentence needs to be quantitative. Also, when calculating variance, remember to use equivalent temporal windows for the Holocene and deglacial and smooth the record to a constant resolution.

Sentence deleted.

5. Discussion
5.1 Sources of brGDGT and implications for CBT/MBT'-derived temperatures Line 325:
Either here or in the methods section some very basic discussion of how to interpret the BIT-values should be given.
Included into section 3.4.

Line 330: Eminent might not be the right word choice for this sentence. Perhaps 'Marine settings where terrigenous input is low are particularly sensitive to bias from in-situ production, thus non-soil derived brGDGTs potentially have a considerable effect on the temperature: : :' Changed.

Line 334: Again, would suggest minor reworking. Perhaps 'Ti/Ca-ratios reflect the proportion of terrigenous and marine derived inorganic components of the sediment, and can be used as an estimator of terrigenous input'. Changed.

Line 336: 'With relatively high values at 15.5 and 12 ka BP, and minima at 14 and 11 ka BP' is an incomplete sentence. Also, again, if presenting chronologies at the centennial scale really need to give errors on those ages. Sentence completed.

Line 362: 'Mai' should be 'May'. Changed.

5.2 Temperature evolution over the past 20 ka 5.2.1 The LGM (20-18 ka) – warm summers and the regional context

What definition of LGM are you using? Should give a reference. Clark et al., 2009 is the most widely used citation that I'm aware of and they define global LGM as ending at 19 ka. We are referring the definition of Mix et al. (2001) according to which the LGM lasted from 18-24 ka BP. This definition is given in the introduction of the revised manuscript (section 1).

Line 371: While you could say there was a 'cooling tendency' from MIS-3 into the LGM, since time moves forward when comparing the LGM to the Holocene it would be better to say 'Generally cooler LGM temperatures are thought to result from: : :' That is a good point, thank you. Sentence is rearranged.

Line 378: What does 'BLB' stand for? BLB stands for "Bering Land Bridge". The abbreviation is removed everywhere in the text.

Line 382: No need to hyphenate 'insect-data'. Also suggest rewording to 'Markovo, and ElGygytgyn and Jack London lakes' Changed.

**5.2.2. Controls on MATifs**

In this section you identify a possible seasonal bias in alkenone-based SST reconstructions towards warmer temperatures and dismiss them in favor of TEXL86 reconstructions. You then discuss the results of the TEXL86 reconstruction for site 12KL currently submitted for review. However there is no discussion of the already-published alkenone-based SST record for 12KL of Max et al., (2012), nor is there a presentation of this record alongside the

TEXL86 record from the same site in Figure 2. For the period of overlap, it would appear that at least at this location the alkenone SST's are several degrees colder than the TEXL86 temperature reconstruction. Why would this be?

The alkenone temperature record from site 12KL is excluded from the LGM discussion since it does not reach beyond 16 ka BP (see Max et al., 2012). For the discussion on millennial scale oscillations we did not show the  $U^{K'_{37}}$  as it is in line with the TEXL86 from core 12KL (Meyer et al., 2016). Since the general trend is the same in both records it appears more reasonable to represent the SST evolution of the NW Pacific by the TEX which spans the entire LGM-Holocene transition.

Differences between UK'37 and TEXL86 are discussed in Meyer et al., 2016 (in the paper referenced as "Meyer et al., submitted" in the original version of the manuscript) and are attributed to different blooming seasons of coccolithophores and archaea.

Line 390: Need to clarify that you're discussing warm Siberian summers during LGM Changed.

Line 414: As the paper Meyer et al., (submitted) has yet to pass through peer review, probably best to state that the relatively warm SST's at site 12KL may be explained by stronger-than-present influence of the Alaskan Stream.

Since the Meyer et al (submitted) is now accepted (Meyer et al., 2016), nothing is changed.

**5.2.3 The deglaciation (18 ka-10 ka BP)**

Define/defend the use of the words 'strong' and 'clear' when describing the resemblance between the N-Atlantic d18O and 12KL MAT. Can you calculate covariance between the normalized/equivalently smoothed NGRIP d18O and 12KL MAT? To my eye they appear quite different: the Y-D is greatly compressed in the Kamchatka MAT record, the trend from the LGM to HS1 in 12KL is completely absent in NGRIP. A climate oscillation in HS1 apparently comparable in magnitude and duration to the regional expression of the Y-D (although a warming as opposed to cooling event) at 16 ka with no analogue in NGRIP is discarded from interpretation. I'm not arguing that there are similarities, but to say it's obvious or 'undoubtable' that the North Atlantic is driving NW Pacific climate via atmospheric teleconnection is a strong claim that needs to be quantitatively defensible. If this can't be done in the context of this paper, perhaps dial the tone of the text down a bit. Also, as stated earlier in the text, when comparing 12KL to NGRIP at centennial scales chronological uncertainties in 12KL need to be addressed and stated. Expressions like "clear" and "strong" and "undoubtedly" are deleted As for age uncertainties, see comment on line 283.

**5.2.4 The Holocene**

The statement at line 620: "Hence it seems that the atmospheric linkage (with the N Atlantic) that determined climate variability during the deglaciation likely persisted into the Holocene where it acted as an important driver for long-term climate changes as well as abrupt, short-lived climate events." seems poorly defended by the visual similarity between NGRIP d180 and Kamchatka MAT in Figure 2. To my eye the Holocene in the MAT record appears more variable, while the mid-Holocene thermal maximum and neoglacial cooling described for the NW Pacific region are absent in NGRIP. Quantitatively evaluating the covariance between these records would be challenging at best as the current chronology for 12KL is virtually

unconstrained in the Holocene. If this statement remains in the discussion/conclusions, at the very least some discussion of what is meant by 'long-term climate changes' versus 'abrupt, short-lived climatic events'.

The Holocene section is rewritten and shortened. This statement is removed.

6. Summary and Conclusions Line 624-629: This introduction to the conclusions reads awkwardly. The introduction of that chapter is rewritten.

Line 631: Perhaps replace 'likely' with 'may' or 'could' as there is no evaluation of statistical certainty of this hypothesis. Replaced by "may".

Line 645: Again, the use of the word 'obvious' to describe the role of N-Atlantic climate in driving the NW Pacific seems somewhere between bombastic and unfounded. There are some similarities in deglacial climate, there are differences, and as yet these remain poorly quantified in the manuscript.

"Obvious" replaced by "seem to be linked".

Figures

Figure 1: Could some kind of shading be used to more clearly denote Holocene landmasses? With apparently identical solid lines used to denote boundaries of continents, ocean currents, and rivers it's a bit difficult to visually parse.

Holocene landmasses are grey, now. (similar as in Figure 1b)

Figure 2: As this figure includes the TEXL86 SST record from Site 12KL to be published in Meyer et al., submitted, it should probably also include the deglacial alkenone SST record from site 12KL published in Max et al., 2012.

Nothing changed as the alkenone record does not reach into the LGM and as the Meyer et al (submitted) is published by now. (Meyer et al., 2016)

Figure 3: As mentioned in my comments on the results section, I think this figure should be reversed with Figure 2 in its presentation order in the text. Also, instead of giving ages at 4 depths in the core, could a secondary axis with appropriately dilated/ compressed ticks be added for age alongside the depth scale? If this isn't possible, would almost suggest it would be better to present results versus time than versus depth to facilitate comparison to Figure 2. As discussed above (see comment on the result section), we do not change the order. We increase the density of the age scale in Figure 3.

[revised manuscript text omitted]
 large parts of Siberia (Fig. 4b). As for Siberia warm summer conditions were explained by increased continentality as a result of the exposed Siberian, Bering and Chukehi Shelfs during the LGM (refernces). However, Iin small parts of the formerly exposed BLB and the arctic shelves temperatures level or exceed PIconditions (Fig. 4b) in the COSMOS simulation. This would be in agreement with tThe exposure of the Siberian Shelf and the BLBmay also have an effect. However, in the model these anomalies are restricted to a relatively 485 small area and are not comparable with the widespread warming tendencies over Siberia, which are visible in the proxy compilation (Fig. 4b, c). Considering that that these positive anomalies are in the area of influence of the dominant anticyclonic anomalies over North America and the associated easterly to southeasterly winds over south Alaska and the BLB (Fig. 4b), it is more likely that the modeled positive anomalies in SAT over the BLB are associated with the changes in atmospheric circulation, rather than with continentality. As for the model proxy 490 discrepancies regarding Siberian SAT one may speculate that the effect of land mass exposure on temperature is underestimated in the model. 
[revised manuscript text omitted]